# DIFUSCO: Graph-based Diffusion Solvers for Combinatorial Optimization

**Zhiqing Sun**[*]
Language Technologies Institute
Carnegie Mellon University
zhiqings@cs.cmu.edu

**Yiming Yang**
Language Technologies Institute
Carnegie Mellon University
yiming@cs.cmu.edu

## Abstract

Neural network-based Combinatorial Optimization (CO) methods have shown promising results in solving various NP-complete (NPC) problems without relying on hand-crafted domain knowledge. This paper broadens the current scope of neural solvers for NPC problems by introducing a new graph-based diffusion framework, namely DIFUSCO. Our framework casts NPC problems as discrete $\{0, 1\}$-vector optimization problems and leverages graph-based denoising diffusion models to generate high-quality solutions. We investigate two types of diffusion models with Gaussian and Bernoulli noise, respectively, and devise an effective inference schedule to enhance the solution quality. We evaluate our methods on two well-studied NPC combinatorial optimization problems: Traveling Salesman Problem (TSP) and Maximal Independent Set (MIS). Experimental results show that DIFUSCO strongly outperforms the previous state-of-the-art neural solvers, improving the performance gap between ground-truth and neural solvers from 1.76% to **0.46%** on TSP-500, from 2.46% to **1.17%** on TSP-1000, and from 3.19% to **2.58%** on TSP-10000. For the MIS problem, DIFUSCO outperforms the previous state-of-the-art neural solver on the challenging SATLIB benchmark.

## 1 Introduction

Combinatorial Optimization (CO) problems are mathematical problems that involve finding the optimal solution in a discrete space. They are fundamental challenges in computer science, especially the NP-Complete (NPC) class of problems, which are believed to be intractable in polynomial time. Traditionally, NPC solvers rely on integer programming (IP) or hand-crafted heuristics, which demand significant expert efforts to approximate near-optimal solutions [4, 31].

Recent development in deep learning has shown new promise in solving NPC problems. Existing neural CO solvers for NPC problems can be roughly classified into three categories based on how the solutions are generated, i.e., the autoregressive constructive solvers, the non-autoregressive constructive solvers, and the improvement heuristics solvers. Methods in the first category use autoregressive factorization to sequentially grow a valid partial solution [6, 64]. Those methods typically suffer from the costly computation in their sequential decoding parts and hence are difficult to scale up to large problems [27]. Methods in the second category rely on non-autoregressive modeling for scaling up, with a conditional independence assumption among variables as typical [53, 55, 92]. Such an assumption, however, unavoidably limits the capability of those methods to capture the multimodal nature of the problems [57, 33], for example, when multiple optimal solutions exists for the same graph. Methods in the third category (improvement heuristics solvers) use a Markov decision process (MDP) to iteratively refines an existing feasible solution with neural network-guided local operations such as 2-opt [71, 2] and node swap [17, 113]. These methods

---

[*]Our code is available at https://github.com/Edward-Sun/DIFUSCO.

37th Conference on Neural Information Processing Systems (NeurIPS 2023).

have also suffered from the difficulty in scaling up and the latency in inference, partly due to the sparse rewards and sample efficiency issues when learning improvement heuristics in a reinforcement learning (RL) framework [113, 79].

Motivated by the recent remarkable success of diffusion models in probabilistic generation [102, 40, 94, 120, 96], we introduce a novel approach named DIFUSCO, which stands for the graph-based DIFfUsion Solvers for Combinatorial Optimization. To apply the iterative denoising process of diffusion models to graph-based settings, we formulate each NPC problem as to find a $\{0, 1\}$-valued vector that indicates the optimal selection of nodes or edges in a candidate solution for the task. Then we use a message passing-based graph neural network [61, 36, 29, 107] to encode each instance graph and to denoise the corrupted variables. Such a graph-based diffusion model overcomes the limitations of previous neural NPC solvers from a new perspective. Firstly, DIFUSCO can perform inference on all variables in parallel with a few ($\ll N$) denoising steps (Sec. 3.3), avoiding the sequential generation problem of autoregressive constructive solvers. Secondly, DIFUSCO can model a multimodal distribution via iterative refinements, which alleviates the expressiveness limitation of previous non-autoregressive constructive models. Last but not least, DIFUSCO is trained in an efficient and stable manner with supervised denoising (Sec. 3.2), which solves the training scalability issue of RL-based improvement heuristics methods.

We should point out that the idea of utilizing a diffusion-based generative model for NPC problems has been explored recently in the literature. In particular, Graikos et al. [32] proposed an image-based diffusion model to solve Euclidean Traveling Salesman problems by projecting each TSP instance onto a $64 \times 64$ greyscale image space and then using a Convolutional Neural Network (CNN) to generate the predicted solution image. The main difference between such *image-based* diffusion solver and our *graph-based* diffusion solver is that the latter can explicitly model the node/edge selection process via the corresponding random variables, which is a natural design choice for formulating NPC problems (since most of them are defined over a graph), while the former does not support such a desirable formalism. Although graph-based modeling has been employed with both constructive [64] and improvement heuristics [20] solvers, how to use graph-based diffusion models for solving NPC problems has not been studied before, to the best of our knowledge.

We investigate two types of probabilistic diffusion modeling within the DIFUSCO framework: continuous diffusion with Gaussian noise [16] and discrete diffusion with Bernoulli noise [5, 44]. These two types of diffusion models have been applied to image processing but not to NPC problems so far. We systematically compare the two types of modeling and find that discrete diffusion performs better than continuous diffusion by a significant margin (Section 4). We also design an effective inference strategy to enhance the generation quality of the discrete diffusion solvers.

Finally, we demonstrate that a single graph neural network architecture, namely the Anisotropic Graph Neural Network [9, 54], can be used as the backbone network for two different NP-complete combinatorial optimization problems: Traveling Salesman Problem (TSP) and Maximal Independent Set (MIS). Our experimental results show that DIFUSCO outperforms previous probabilistic NPC solvers on benchmark datasets of TSP and MIS problems with various sizes.

## 2 Related Work

### 2.1 Autoregressive Construction Heuristics Solvers

Autoregressive models have achieved state-of-the-art results as constructive heuristic solvers for combinatorial optimization (CO) problems, following the recent success of language modeling in the text generation domain [106, 11]. The first approach proposed by Bello et al. [6] uses a neural network with reinforcement learning to append one new variable to the partial solution at each decoding step until a complete solution is generated. However, autoregressive models [64] face high time and space complexity challenges for large-scale NPC problems due to their sequential generation scheme and quadratic complexity in the self-attention mechanism [106].

### 2.2 Non-autoregressive Construction Heuristics Solvers

Non-autoregressive (or heatmap) constructive heuristics solvers [53, 27, 28, 92] are recently proposed to address this scalability issue by assuming conditional independence among variables in NPC problems, but this assumption limits the ability to capture the multimodal nature [57, 33] of high-

quality solution distributions. Therefore, additional active search [6, 92] or Monte-Carlo Tree Search (MCTS) [27, 98] are needed to further improve the expressive power of the non-autoregressive scheme.

DIFUSCO can be regarded as a member in the non-autoregressive constructive heuristics category and thus can be benefited from heatmap search techniques such as MCTS. But DIFUSCO uses an iterative denoising scheme to generate the final heatmap, which significantly enhances its expressive power compared to previous non-autoregressive methods.

### 2.3 Diffusion Models for Discrete Data

Typical diffusion models [100, 102, 40, 103, 85, 56] operate in the continuous domain, progressively adding Gaussian noise to the clean data in the forward process, and learning to remove noises in the reverse process in a discrete-time framework.

Discrete diffusion models have been proposed for the generation of discrete image bits or texts using binomial noises [100] and multinomial/categorical noises [5, 44]. Recent research has also shown the potential of discrete diffusion models in sound generation [118], protein structure generation [77], molecule generation [108], and better text generation [52, 38].

Another line of work studies diffusion models for discrete data by applying continuous diffusion models with Gaussian noise on the embedding space of discrete data [30, 69, 24], the $\{-1.0, 1.0\}$ real-number vector space [16], and the simplex space [37]. The most relevant work might be Niu et al. [86], which proposed a continuous score-based generative framework for graphs, but they only evaluated simple non-NP-hard CO tasks such as Shortest Path and Maximum Spanning Tree.

## 3 DIFUSCO: Proposed Approach

### 3.1 Problem Definition

Following a conventional notation [88], we define $\mathcal{X}_s = \{0, 1\}^N$ as the space of candidate solutions $\{\mathbf{x}\}$ for a CO problem instance $s$, and $c_s : \mathcal{X}_s \to \mathbb{R}$ as the objective function for solution $\mathbf{x} \in \mathcal{X}_s$:

$$c_s(\mathbf{x}) = \mathrm{cost}(\mathbf{x}, s) + \mathrm{valid}(\mathbf{x}, s). \tag{1}$$

Here $\mathrm{cost}(\cdot)$ is the task-specific cost of a candidate solution (e.g., the tour length in TSP), which is often a simple linear function of $\mathbf{x}$ in most NP-complete problems, and $\mathrm{valid}(\cdot)$ is the validation term that returns 0 for a feasible solution and $+\infty$ for an invalid one. The optimization objective is to find the optimal solution $\mathbf{x}^{s*}$ for a given instance $s$ as:

$$\mathbf{x}^{s*} = \underset{\mathbf{x} \in \mathcal{X}_s}{\mathrm{argmin}}\, c_s(\mathbf{x}). \tag{2}$$

This framework is generically applicable to different NPC problems. For example, for the Traveling Salesman Problem (TSP), $\mathbf{x} \in \{0, 1\}^N$ is the indicator vector for selecting a subset from $N$ edges; the cost of this subset is calculated as: $\mathrm{cost}_{\mathrm{TSP}}(\mathbf{x}, s) = \sum_i x_i \cdot d_i^{(s)}$, where $d_i^{(s)}$ denotes the weight of the $i$-th edge in problem instance $s$, and the $\mathrm{valid}(\cdot)$ part of Formula (1) ensures that $\mathbf{x}$ is a tour that visits each node exactly once and returns to the starting node at the end. For the Maximal Independent Set (MIS) problem, $\mathbf{x} \in \{0, 1\}^N$ is the indicator vector for selecting a subset from $N$ nodes; the cost of the subset is calculated as: $\mathrm{cost}_{\mathrm{MIS}}(\mathbf{x}, s) = \sum_i (1 - x_i)$,, and the corresponding $\mathrm{valid}(\cdot)$ validates $\mathbf{x}$ is an independent set where each node in the set has no connection to any other node in the set.

Probabilistic neural NPC solvers [6] tackle instance problem $s$ by defining a parameterized conditional distribution $p_{\boldsymbol{\theta}}(\mathbf{x}|s)$, such that the expected cost $\sum_{\mathbf{x} \in \mathcal{X}_s} c_s(\mathbf{x}) \cdot p(\mathbf{x}|s)$ is minimized. Such probabilistic generative models are usually optimized by reinforcement learning algorithms [111, 63]. In this paper, assuming the optimal (or high-quality) solution $\mathbf{x}_s^*$ is available for each training instance $s$, we optimize the model through supervised learning. Let $\mathcal{S} = \{s_i\}_1^N$ be independently and identically distributed (IID) training samples for a type of NPC problem, we aim to maximize the likelihood of optimal (or high-quality) solutions, where the loss function $L$ is defined as:

$$L(\boldsymbol{\theta}) = \mathbb{E}_{s \in \mathcal{S}}\left[-\log p_{\boldsymbol{\theta}}(\mathbf{x}^{s*}|s)\right] \tag{3}$$

Next, we describe how to use diffusion models to parameterize the generative distribution $p_{\boldsymbol{\theta}}$. For brevity, we omit the conditional notations of $s$ and denote $\mathbf{x}^{s*}$ as $\mathbf{x}_0$ as a convention in diffusion models for all formulas in the the rest of the paper.

## 3.2 Diffusion Models in DIFUSCO

From the variational inference perspective [60], diffusion models [100, 40, 102] are latent variable models of the form $p_{\boldsymbol{\theta}}(\mathbf{x}_0) := \int p_{\boldsymbol{\theta}}(\mathbf{x}_{0:T}) d\mathbf{x}_{1:T}$, where $\mathbf{x}_1, \ldots, \mathbf{x}_T$ are latents of the same dimensionality as the data $\mathbf{x}_0 \sim q(\mathbf{x}_0)$. The joint distribution $p_{\boldsymbol{\theta}}(\mathbf{x}_{0:T}) = p(\mathbf{x}_T) \prod_{t=1}^{T} p_{\boldsymbol{\theta}}(\mathbf{x}_{t-1}|\mathbf{x}_t)$ is the learned reverse (denoising) process that gradually denoises the latent variables toward the data distribution, while the forward process $q(\mathbf{x}_{1:T}|\mathbf{x}_0) = \prod_{t=1}^{T} q(\mathbf{x}_t|\mathbf{x}_{t-1})$ gradually corrupts the data into noised latent variables. Training is performed by optimizing the usual variational bound on negative log-likelihood:

$$
\begin{aligned}
\mathbb{E}\left[-\log p_{\boldsymbol{\theta}}(\mathbf{x}_0)\right] &\le \mathbb{E}_q\left[-\log \frac{p_{\boldsymbol{\theta}}(\mathbf{x}_{0:T})}{q_{\boldsymbol{\theta}}(\mathbf{x}_{1:T}|\mathbf{x}_0)}\right] \\
&= \mathbb{E}_q\left[\sum_{t>1} D_{KL}[q(\mathbf{x}_{t-1}|\mathbf{x}_t, \mathbf{x}_0)\|p_{\boldsymbol{\theta}}(\mathbf{x}_{t-1}|\mathbf{x}_t)] - \log p_{\boldsymbol{\theta}}(\mathbf{x}_0|\mathbf{x}_1)\right] + C
\end{aligned}
\tag{4}
$$

where $C$ is a constant.

**Discrete Diffusion** In discrete diffusion models with multinomial noises [5, 44], the forward process is defined as: $q(\mathbf{x}_t|\mathbf{x}_{t-1}) = \mathrm{Cat}\left(\mathbf{x}_t; \mathbf{p} = \tilde{\mathbf{x}}_{t-1}\mathbf{Q}_t\right)$, where $\mathbf{Q}_t = \begin{bmatrix} (1-\beta_t) & \beta_t \\ \beta_t & (1-\beta_t) \end{bmatrix}$ is the transition probability matrix; $\tilde{\mathbf{x}} \in \{0,1\}^{N \times 2}$ is converted from the original vector $\mathbf{x} \in \{0,1\}^N$ with a one-hot vector per row; and $\tilde{\mathbf{x}}\mathbf{Q}$ computes a row-wise vector-matrix product.

Here, $\beta_t$ denotes the corruption ratio. Also, we want $\prod_{t=1}^{T}(1-\beta_t) \approx 0$ such that $\mathbf{x}_T \sim \mathrm{Uniform}(\cdot)$. The $t$-step marginal can thus be written as: $q(\mathbf{x}_t|\mathbf{x}_0) = \mathrm{Cat}\left(\mathbf{x}_t; \mathbf{p} = \tilde{\mathbf{x}}_0\overline{\mathbf{Q}}_t\right)$, where $\overline{\mathbf{Q}}_t = \mathbf{Q}_1\mathbf{Q}_2 \ldots \mathbf{Q}_t$. And the posterior at time $t-1$ can be obtained by Bayes' theorem:

$$
q(\mathbf{x}_{t-1}|\mathbf{x}_t, \mathbf{x}_0) = \frac{q(\mathbf{x}_t|\mathbf{x}_{t-1}, \mathbf{x}_0)q(\mathbf{x}_{t-1}|\mathbf{x}_0)}{q(\mathbf{x}_t|\mathbf{x}_0)} = \mathrm{Cat}\left(\mathbf{x}_{t-1}; \mathbf{p} = \frac{\tilde{\mathbf{x}}_t\mathbf{Q}_t^{\top} \odot \tilde{\mathbf{x}}_0\overline{\mathbf{Q}}_{t-1}}{\tilde{\mathbf{x}}_0\overline{\mathbf{Q}}_t\tilde{\mathbf{x}}_t^{\top}}\right), \tag{5}
$$

where $\odot$ denotes the element-wise multiplication.

According to Austin et al. [5], the denoising neural network is trained to predict the clean data $p_{\boldsymbol{\theta}}(\tilde{\mathbf{x}}_0|\mathbf{x}_t)$, and the reverse process is obtained by substituting the predicted $\tilde{\mathbf{x}}_0$ as $\mathbf{x}_0$ in Eq. 5:

$$
p_{\boldsymbol{\theta}}(\mathbf{x}_{t-1}|\mathbf{x}_t) = \sum_{\tilde{\mathbf{x}}} q(\mathbf{x}_{t-1}|\mathbf{x}_t, \tilde{\mathbf{x}}_0)p_{\boldsymbol{\theta}}(\tilde{\mathbf{x}}_0|\mathbf{x}_t) \tag{6}
$$

**Continuous Diffusion for Discrete Data** The continuous diffusion models [102, 40] can also be directly applied to discrete data by lifting the discrete input into a continuous space [16]. Since the continuous diffusion models usually start from a standard Gaussian distribution $\boldsymbol{\epsilon} \sim \mathcal{N}(\mathbf{0}, \mathbf{I})$, Chen et al. [16] proposed to first rescale the $\{0,1\}$-valued variables $\mathbf{x}_0$ to the $\{-1, 1\}$ domain as $\hat{\mathbf{x}}_0$, and then treat them as real values. The forward process in continuous diffusion is defined as: $q(\hat{\mathbf{x}}_t|\hat{\mathbf{x}}_{t-1}) := \mathcal{N}(\hat{\mathbf{x}}_t; \sqrt{1-\beta_t}\hat{\mathbf{x}}_{t-1}, \beta_t\mathbf{I})$.

Again, $\beta_t$ denotes the corruption ratio, and we want $\prod_{t=1}^{T}(1-\beta_t) \approx 0$ such that $\mathbf{x}_T \sim \mathcal{N}(\cdot)$. The $t$-step marginal can thus be written as: $q(\hat{\mathbf{x}}_t|\hat{\mathbf{x}}_0) := \mathcal{N}(\hat{\mathbf{x}}_t; \sqrt{\bar{\alpha}_t}\hat{\mathbf{x}}_0, (1-\bar{\alpha}_t)\mathbf{I})$ where $\alpha_t = 1 - \beta_t$ and $\bar{\alpha}_t = \prod_{\tau=1}^{t} \alpha_\tau$. Similar to Eq. 5, the posterior at time $t-1$ can be obtained by Bayes' theorem:

$$
q(\hat{\mathbf{x}}_{t-1}|\hat{\mathbf{x}}_t, \mathbf{x}_0) = \frac{q(\hat{\mathbf{x}}_t|\hat{\mathbf{x}}_{t-1}, \hat{\mathbf{x}}_0)q(\hat{\mathbf{x}}_{t-1}|\hat{\mathbf{x}}_0)}{q(\hat{\mathbf{x}}_t|\hat{\mathbf{x}}_0)}, \tag{7}
$$

which is a closed-form Gaussian distribution [40]. In continuous diffusion, the denoising neural network is trained to predict the unscaled Gaussian noise $\tilde{\boldsymbol{\epsilon}}_t = (\hat{\mathbf{x}}_t - \sqrt{\bar{\alpha}_t}\hat{\mathbf{x}}_0)/\sqrt{1-\bar{\alpha}_t} = f_{\boldsymbol{\theta}}(\hat{\mathbf{x}}_t, t)$. The reverse process [40] can use a point estimation of $\hat{\mathbf{x}}_0$ in the posterior:

$$
p_{\boldsymbol{\theta}}(\hat{\mathbf{x}}_{t-1}|\hat{\mathbf{x}}_t) = q\left(\hat{\mathbf{x}}_{t-1}|\hat{\mathbf{x}}_t, \frac{\hat{\mathbf{x}}_t - \sqrt{1-\bar{\alpha}_t}f_{\boldsymbol{\theta}}(\hat{\mathbf{x}}_t, t)}{\sqrt{\bar{\alpha}_t}}\right) \tag{8}
$$

For generating discrete data, after the continuous data $\hat{\mathbf{x}}_0$ is generated, a thresholding/quantization operation is applied to convert them back to $\{0,1\}$-valued variables $\mathbf{x}_0$ as the model's prediction.

### 3.3 Denoising Schedule for Fast Inference

One way to speed up the inference of denoising diffusion models is to reduce the number of steps in the reverse diffusion process, which also reduces the number of neural network evaluations. The denoising diffusion implicit models (DDIMs) [101] are a class of models that apply this strategy in the continuous domain, and a similar approach can be used for discrete diffusion models [5].

Formally, when the forward process is defined not on all the latent variables $\mathbf{x}_{1:T}$, but on a subset $\{\mathbf{x}_{\tau_1}, \ldots, \mathbf{x}_{\tau_M}\}$, where $\tau$ is an increasing sub-sequence of $[1, \ldots, T]$ with length $M$, $\mathbf{x}_{\tau_1} = 1$ and $\mathbf{x}_{\tau_M} = T$, the fast sampling algorithms directly models $q(\mathbf{x}_{\tau_{i-1}}|\mathbf{x}_{\tau_i}, \mathbf{x}_0)$. Due to the space limit, the detailed algorithms are described in the appendix.

We consider two types of denoising scheduled for $\tau$ given the desired $\text{card}(\tau) < T$: `linear` and `cosine`. The former uses timesteps such that $\tau_i = \lfloor ci \rfloor$ for some $c$, and the latter uses timesteps such that $\tau_i = \lfloor \cos(\frac{(1-ci)\pi}{2}) \cdot T \rfloor$ for some $c$. The intuition for the cosine schedule is that diffusion models can achieve better generation quality when iterating more steps in the low-noise regime [85, 121, 13].

### 3.4 Graph-based Denoising Network

The denoising network takes as input a set of noisy variables $\mathbf{x}_t$ and the problem instance $s$ and predicts the clean data $\widetilde{\mathbf{x}}_0$. To balance both scalability and performance considerations, we adopt an anisotropic graph neural network with edge gating mechanisms [9, 54] as the backbone network for both discrete and continuous diffusion models, and the variables in the network output can be the states of either nodes, as in the case of Maximum Independent Set (MIS) problems, or edges, as in the case of Traveling Salesman Problems (TSP). Our choice of network mainly follows previous work [54, 92], as AGNN can produce the embeddings for both nodes and edges, unlike typical GNNs such as GCN [62] or GAT [107], which are designed for node embedding only. This design choice is particularly beneficial for tasks that require the prediction of edge variables.

**Anisotropic Graph Neural Networks** Let $\boldsymbol{h}_i^\ell$ and $\boldsymbol{e}_{ij}^\ell$ denote the node and edge features at layer $\ell$ associated with node $i$ and edge $ij$, respectively. $\mathbf{t}$ is the sinusoidal features [106] of denoising timestep $t$. The features at the next layer is propagated with an anisotropic message passing scheme:

$$\hat{\boldsymbol{e}}_{ij}^{\ell+1} = \boldsymbol{P}^\ell \boldsymbol{e}_{ij}^\ell + \boldsymbol{Q}^\ell \boldsymbol{h}_i^\ell + \boldsymbol{R}^\ell \boldsymbol{h}_j^\ell,$$
$$\boldsymbol{e}_{ij}^{\ell+1} = \boldsymbol{e}_{ij}^\ell + \text{MLP}_e(\text{BN}(\hat{\boldsymbol{e}}_{ij}^{\ell+1})) + \text{MLP}_t(\mathbf{t}),$$
$$\boldsymbol{h}_i^{\ell+1} = \boldsymbol{h}_i^\ell + \alpha(\text{BN}(\boldsymbol{U}^\ell \boldsymbol{h}_i^\ell + \mathcal{A}_{j \in \mathcal{N}_i}(\sigma(\hat{\boldsymbol{e}}_{ij}^{\ell+1}) \odot \boldsymbol{V}^\ell \boldsymbol{h}_j^\ell))),$$

where $\boldsymbol{U}^\ell, \boldsymbol{V}^\ell, \boldsymbol{P}^\ell, \boldsymbol{Q}^\ell, \boldsymbol{R}^\ell \in \mathbb{R}^{d \times d}$ are the learnable parameters of layer $\ell$, $\alpha$ denotes the ReLU [66] activation, BN denotes the Batch Normalization operator [51], $\mathcal{A}$ denotes the aggregation function SUM pooling [116], $\sigma$ is the sigmoid function, $\odot$ is the Hadamard product, $\mathcal{N}_i$ denotes the neighborhoods of node $i$, and $\text{MLP}_{(.)}$ denotes a 2-layer multi-layer perceptron.

For TSP, $\boldsymbol{e}_{ij}^0$ are initialized as the corresponding values in $\mathbf{x}_t$, and $\boldsymbol{h}_i^0$ are initialized as sinusoidal features of the nodes. For MIS, $\boldsymbol{e}_{ij}^0$ are initialized as zeros, and $\boldsymbol{h}_i^0$ are initialized as the corresponding values in $\mathbf{x}_t$. A 2-neuron and 1-neuron classification/regression head is applied to the final embeddings of $\mathbf{x}_t$ ($\{\mathbf{e}_{ij}\}$ for edges and $\{\mathbf{h}_i\}$ for nodes) for discrete and continuous diffusion models, respectively.

**Hyper-parameters** For all TSP and MIS benchmarks, we use a 12-layer Anisotropic GNN with a width of 256 as described above.

### 3.5 Decoding Strategies for Diffusion-based Solvers

After the training of the parameterized denoising network according to Eq. 4, the solutions are sampled from the diffusion models $p_{\boldsymbol{\theta}}(\mathbf{x}_0|s)$ for final evaluation. However, probabilistic generative models such as DIFUSCO cannot guarantee that the sampled solutions are feasible according to the definition of CO problems. Therefore, specialized decoding strategies are designed for the two CO problems studied in this paper.

**Heatmap Generation**  The diffusion models $p_{\boldsymbol{\theta}}(\cdot|s)$ produce discrete variables $\mathbf{x}$ as the final predictions by applying Bernoulli sampling (Eq. 6) for discrete diffusion or quantization for continuous diffusion. However, this process discards the comparative information that reflects the confidence of the predicted variables, which is crucial for resolving conflicts in the decoding process. To preserve this information, we adapt the diffusion models to generate heatmaps [53, 92] by making the following appropriate modifications: 1) For discrete diffusion, the final score of $p_{\boldsymbol{\theta}}(\mathbf{x_0} = 1|s)$ is preserved as the heatmap scores; 2) For continuous diffusion, we remove the final quantization and use $0.5(\hat{\mathbf{x}}_0 + 1)$ as the heatmap scores. Note that different from previous heatmap approaches [53, 92] that produce a single conditionally independent distribution for all variables, DIFUSCO can produce diverse multimodal output distribution by using different random seeds.

**TSP Decoding**  Let $\{A_{ij}\}$ be the heatmap scores generated by DIFUSCO denoting the confidence of each edge. We evaluate two approaches as the decoding method following previous work [32, 92]: 1) Greedy decoding [32], where all the edges are ranked by $(A_{ij} + A_{ji})/\|\mathbf{c}_i - \mathbf{c}_j\|$, and are inserted into the partial solution if there are no conflicts. 2-opt heuristics [71] are optionally applied. 2) Monte Carlo Tree Search (MCTS) [27], where $k$-opt transformation actions are sampled guided by the heatmap scores to improve the current solutions. Due to the space limit, a detailed description of two decoding strategies can be found in the appendix.

**MIS Decoding**  Let $\{a_i\}$ be the heatmap scores generated by DIFUSCO denoting the confidence of each node. A greedy decoding strategy is used for the MIS problem, where the nodes are ranked by $a_i$ and inserted into the partial solution if there are no conflicts. Recent research [8] pointed out that the graph reduction and 2-opt search [2] can find near-optimal solutions even starting from a randomly generated solution, so we do not use any post-processing for the greedy-decoded solutions.

**Solution Sampling**  A common practice for probabilistic CO solvers [64] is to sample multiple solutions and report the best one. For DIFUSCO, we follow this practice by sampling multiple heatmaps from $p_{\boldsymbol{\theta}}(\mathbf{x}_0|s)$ with different random seeds and then applying the greedy decoding algorithm described above to each heatmap.

# 4 Experiments with TSP

We use 2-D Euclidean TSP instances to test our models. We generate these instances by randomly sampling nodes from a uniform distribution over the unit square. We use TSP-50 (with 50 nodes) as the main benchmark to compare different model configurations. We also evaluate our method on larger TSP instances with 100, 500, 1000, and 10000 nodes to demonstrate its scalability and performance against other state-of-the-art methods.

## 4.1 Experimental Settings

**Datasets**  We generate and label the training instances using the Concorde exact solver [3] for TSP-50/100 and the LKH-3 heuristic solver [39] for TSP-500/1000/10000. We use the same test instances as [54, 64] for TSP-50/100 and [27] for TSP-500/1000/10000.

**Graph Sparsification**  We use sparse graphs for large-scale TSP problems to reduce the computational complexity. We sparsify the graphs by limiting each node to have only $k$ edges to its nearest neighbors based on the Euclidean distances. We set $k$ to 50 for TSP-500 and 100 for TSP-1000/10000. This way, we avoid the quadratic growth of edges in dense graphs as the number of nodes increases.

**Model Settings**  $T = 1000$ denoising steps are used for the training of DIFUSCO on all datasets. Following Ho et al. [40], Graikos et al. [32], we use a simple linear noise schedule for $\{\beta_t\}_{t=1}^T$, where $\beta_1 = 10^{-4}$ and $\beta_T = 0.02$. We follow Graikos et al. [32] and use the Greedy decoding + 2-opt scheme (Sec. 3.5) as the default decoding scheme for experiments.

**Evaluation Metrics**  In order to compare the performance of different models, we present three metrics: average tour length (Length), average relative performance gap (Gap), and total run time (Time). The detailed description can be found in the appendix.

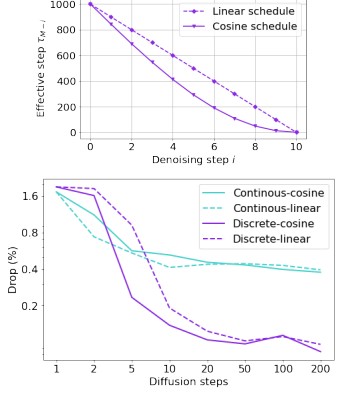

Figure 1: Comparison of continuous (Gaussian noise) and discrete (Bernoulli noise) diffusion models with different inference diffusion steps and inference schedule (`linear` v.s. `cosine`).

Table 1: Comparing results on TSP-50 and TSP-100. ∗ denotes the baseline for computing the performance gap. † indicates that the diffusion model samples a single solution as its greedy decoding scheme. Please refer to Sec. 4 for details.

| ALGORITHM | TYPE | TSP-50 LENGTH↓ | TSP-50 GAP(%)↓ | TSP-100 LENGTH↓ | TSP-100 GAP(%)↓ |
|---|---|---|---|---|---|
| CONCORDE* | EXACT | 5.69 | 0.00 | 7.76 | 0.00 |
| 2-OPT | HEURISTICS | 5.86 | 2.95 | 8.03 | 3.54 |
| AM | GREEDY | 5.80 | 1.76 | 8.12 | 4.53 |
| GCN | GREEDY | 5.87 | 3.10 | 8.41 | 8.38 |
| TRANSFORMER | GREEDY | 5.71 | 0.31 | 7.88 | 1.42 |
| POMO | GREEDY | 5.73 | 0.64 | 7.84 | 1.07 |
| SYM-NCO | GREEDY | - | - | 7.84 | 0.94 |
| DPDP | $1k$-IMPROVEMENTS | 5.70 | 0.14 | 7.89 | 1.62 |
| IMAGE DIFFUSION | GREEDY† | 5.76 | 1.23 | 7.92 | 2.11 |
| **OURS** | GREEDY† | **5.70** | **0.10** | **7.78** | **0.24** |
| AM | $1k\times$SAMPLING | 5.73 | 0.52 | 7.94 | 2.26 |
| GCN | $2k\times$SAMPLING | 5.70 | 0.01 | 7.87 | 1.39 |
| TRANSFORMER | $2k\times$SAMPLING | 5.69 | 0.00 | 7.76 | 0.39 |
| POMO | $8\times$AUGMENT | 5.69 | 0.03 | 7.77 | 0.14 |
| SYM-NCO | $100\times$SAMPLING | - | - | 7.79 | 0.39 |
| MDAM | $50\times$SAMPLING | 5.70 | 0.03 | 7.79 | 0.38 |
| DPDP | $100k$-IMPROVEMENTS | 5.70 | 0.00 | 7.77 | 0.00 |
| **OURS** | $16\times$SAMPLING | **5.69** | **-0.01** | **7.76** | **-0.01** |

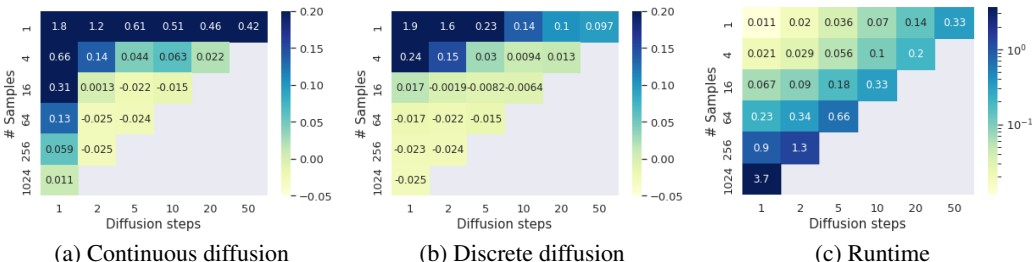

(a) Continuous diffusion     (b) Discrete diffusion     (c) Runtime

Figure 2: The performance Gap (%) are shown for continuous diffusion **(a)** and discrete diffusion **(b)** models on TSP-50 with different diffusion steps and number of samples. Their corresponding per-instance run-time (sec) are shown in **(c)**, where the decomposed analysis (neural network + greedy decoding + 2-opt) can be found in the appendix.

## 4.2 Design Analysis

**Discrete Diffusion v.s. Continuous Diffusion**   We first investigate the suitability of two diffusion approaches for combinatorial optimization, namely continuous diffusion with Gaussian noise and discrete diffusion with Bernoulli noise (Sec. 3.2). Additionally, we explore the effectiveness of different denoising schedules, such as `linear` and `cosine` schedules (Sec. 3.3), on CO problems. To efficiently evaluate these model choices, we utilize the TSP-50 benchmark.

Note that although all the diffusion models are trained with a $T = 1000$ noise schedule, the inference schedule can be shorter than $T$, as described in Sec. 3.3. Specifically, we are interested in diffusion models with 1, 2, 5, 10, 20, 50, 100, and 200 diffusion steps.

Fig. 1 demonstrates the performance of two types of diffusion models with two types of inference schedules and various diffusion steps. We can see that discrete diffusion consistently outperforms the continuous diffusion models by a large margin when there are more than 5 diffusion steps[2]. Besides, the `cosine` schedule is superior to `linear` on discrete diffusion and performs similarly on continuous diffusion. Therefore, we use `cosine` for the rest of the paper.

**More Diffusion Iterations v.s. More Sampling**   By utilizing effective denoising schedules, diffusion models are able to adaptively infer based on the available computation budget by predetermining the total number of diffusion steps. This is similar to changing the number of samples in previous probabilistic neural NPC solvers [64]. Therefore, we investigate the trade-off between the number of diffusion iterations and the number of samples for diffusion-based NPC solvers.

---

[2]We also observe similar patterns on TSP-100, where the results are reported in the appendix.

Table 2: Results on large-scale TSP problems. RL, SL, AS, G, S, BS, and MCTS denotes Reinforcement Learning, Supervised Learning, Active Search, Greedy decoding, Sampling decoding, Beam-search, and Monte Carlo Tree Search, respectively. * indicates the baseline for computing the performance gap. Results of baselines are taken from Fu et al. [27] and Qiu et al. [92], so the runtime may not be directly comparable. See Section 4 and appendix for detailed descriptions.

| ALGORITHM | TYPE | TSP-500 | | | TSP-1000 | | | TSP-10000 | | |
|---|---|---|---|---|---|---|---|---|---|---|
| | | LENGTH↓ | GAP↓ | TIME↓ | LENGTH↓ | GAP↓ | TIME↓ | LENGTH↓ | GAP↓ | TIME↓ |
| CONCORDE | EXACT | 16.55* | — | 37.66m | 23.12* | — | 6.65h | N/A | N/A | N/A |
| GUROBI | EXACT | 16.55 | 0.00% | 45.63h | N/A | N/A | N/A | N/A | N/A | N/A |
| LKH-3 (DEFAULT) | HEURISTICS | 16.55 | 0.00% | 46.28m | 23.12 | 0.00% | 2.57h | 71.77* | — | 8.8h |
| LKH-3 (LESS TRAILS) | HEURISTICS | 16.55 | 0.00% | 3.03m | 23.12 | 0.00% | 7.73h | 71.79 | — | 51.27m |
| FARTHEST INSERTION | HEURISTICS | 18.30 | 10.57% | 0s | 25.72 | 11.25% | 0s | 80.59 | 12.29% | 6s |
| AM | RL+G | 20.02 | 20.99% | 1.51m | 31.15 | 34.75% | 3.18m | 141.68 | 97.39% | 5.99m |
| GCN | SL+G | 29.72 | 79.61% | 6.67m | 48.62 | 110.29% | 28.52m | N/A | N/A | N/A |
| POMO+EAS-EMB | RL+AS+G | 19.24 | 16.25% | 12.80h | N/A | N/A | N/A | N/A | N/A | N/A |
| POMO+EAS-TAB | RL+AS+G | 24.54 | 48.22% | 11.61h | 49.56 | 114.36% | 63.45h | N/A | N/A | N/A |
| DIMES | RL+G | 18.93 | 14.38% | 0.97m | 26.58 | 14.97% | 2.08m | 86.44 | 20.44% | 4.65m |
| DIMES | RL+AS+G | 17.81 | 7.61% | 2.10h | 24.91 | 7.74% | 4.49h | 80.45 | 12.09% | 3.07h |
| OURS (DIFUSCO) | SL+G† | 18.35 | 10.85% | 3.61m | 26.14 | 13.06% | 11.86m | 98.15 | 36.75% | 28.51m |
| OURS (DIFUSCO) | SL+G†+2-OPT | 16.80 | 1.49% | 3.65m | 23.56 | 1.90% | 12.06m | 73.99 | 3.10% | 35.38m |
| EAN | RL+S+2-OPT | 23.75 | 43.57% | 57.76m | 47.73 | 106.46% | 5.39h | N/A | N/A | N/A |
| AM | RL+BS | 19.53 | 18.03% | 21.99m | 29.90 | 29.23% | 1.64h | 129.40 | 80.28% | 1.81h |
| GCN | SL+BS | 30.37 | 83.55% | 38.02m | 51.26 | 121.73% | 51.67m | N/A | N/A | N/A |
| DIMES | RL+S | 18.84 | 13.84% | 1.06m | 26.36 | 14.01% | 2.38m | 85.75 | 19.48% | 4.80m |
| DIMES | RL+AS+S | 17.80 | 7.55% | 2.11h | 24.89 | 7.70% | 4.53h | 80.42 | 12.05% | 3.12h |
| OURS (DIFUSCO) | SL+S | 17.23 | 4.08% | 11.02m | 25.19 | 8.95% | 46.08m | 95.52 | 33.09% | 6.59h |
| OURS (DIFUSCO) | SL+S+2-OPT | 16.65 | 0.57% | 11.46m | 23.45 | 1.43% | 48.09m | 73.89 | 2.95% | 6.72h |
| ATT-GCN | SL+MCTS | 16.97 | 2.54% | 2.20m | 23.86 | 3.22% | 4.10m | 74.93 | 4.39% | 21.49m |
| DIMES | RL+MCTS | 16.87 | 1.93% | 2.92m | 23.73 | 2.64% | 6.87m | 74.63 | 3.98% | 29.83m |
| DIMES | RL+AS+MCTS | 16.84 | 1.76% | 2.15h | 23.69 | 2.46% | 4.62h | 74.06 | 3.19% | 3.57h |
| OURS (DIFUSCO) | SL+MCTS | 16.63 | 0.46% | 10.13m | 23.39 | 1.17% | 24.47m | 73.62 | 2.58% | 47.36m |

Fig. 2 shows the results of continuous diffusion and discrete diffusion with various diffusion steps and number of parallel sampling, as well as their corresponding total run-time. The `cosine` denoising schedule is used for fast inference. Again, we find that discrete diffusion outperforms continuous diffusion across various settings. Besides, we find performing more diffusion iterations is generally more effective than sampling more solutions, even when the former uses less computation. For example, 20 (diffusion steps) $\times$ 4 (samples) performs competitive to 1 (diffusion steps) $\times$ 1024 (samples), while the runtime of the former is $18.5\times$ less than the latter.

In general, we find that 50 (diffusion steps) $\times$ 1 (samples) policy and 10 (diffusion steps) $\times$ 16 (samples) policy make a good balance between exploration and exploitation for discrete DIFUSCO models and use them as the **Greedy** and **Sampling** strategies for the rest of the experiments.

## 4.3 Main Results

**Comparison to SOTA Methods** We compare discrete DIFUSCO to other state-of-the-art neural NPC solvers on TSP problems across various scales. Due to the space limit, the description of other baseline models can be found in the appendix.

Tab. 1 compare discrete DIFUSCO with other models on TSP-50 and TSP-100, where DIFUSCO achieves the state-of-the-art performance in both greedy and sampling settings for probabilistic solvers.

Tab. 2 compare discrete DIFUSCO with other models on the larger-scale TSP-500, TSP-1000, and TSP-10000 problems. Most previous probabilistic solvers (except DIMES [92]) becomes untrainable on TSP problems of these scales, so the results of these models are reported with TSP-100 trained models. The results are reported with greedy, sampling, and MCTS decoding strategies, respectively. For fair comparisons [27, 92], MCTS decoding for TSP is always evaluated with only one sampled heatmap. From the table, we can see that DIFUSCO strongly outperforms the previous neural solvers on all three settings. In particular, with MCTS-based decoding, DIFUSCO significantly improving the performance gap between ground-truth and neural solvers from 1.76% to **0.46%** on TSP-500, from 2.46% to **1.17%** on TSP-1000, and from 3.19% to **2.58%** on TSP-10000.

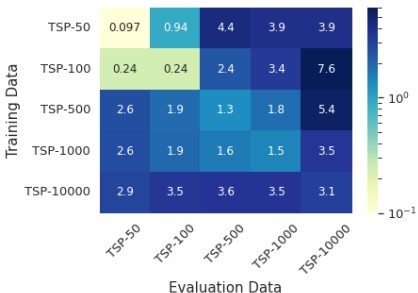

Figure 3: Generalization tests of DIFUSCO trained and evaluated on TSP problems across various scales. The performance Gap (%) with greedy decoding and 2-opt is reported.

Table 3: Results on MIS problems. * indicates the baseline for computing the optimality gap. RL, SL, G, S, and TS denote Reinforcement Learning, Supervised Learning, Greedy decoding, Sampling decoding, and Tree Search, respectively. Please refer to Sec. 5 and appendix for details.

| METHOD | TYPE | SATLIB | | | ER-[700-800] | | |
| | | SIZE ↑ | GAP ↓ | TIME ↓ | SIZE ↑ | GAP ↓ | TIME ↓ |
|---|---|---|---|---|---|---|---|
| KaMIS | HEURISTICS | 425.96* | — | 37.58m | 44.87* | — | 52.13m |
| GUROBI | EXACT | 425.95 | 0.00% | 26.00m | 41.38 | 7.78% | 50.00m |
| INTEL | SL+G | 420.66 | 1.48% | 23.05m | 34.86 | 22.31% | 6.06m |
| INTEL | SL+TS | N/A | N/A | N/A | 38.80 | 13.43% | 20.00M |
| DGL | SL+TS | N/A | N/A | N/A | 37.26 | 16.96% | 22.71m |
| LwD | RL+S | 422.22 | 0.88% | 18.83m | 41.17 | 8.25% | 6.33m |
| DIMES | RL+G | 421.24 | 1.11% | 24.17m | 38.24 | 14.78% | 6.12m |
| DIMES | RL+S | 423.28 | 0.63% | 20.26m | **42.06** | **6.26%** | 12.01m |
| OURS | SL+G | **424.50** | **0.34%** | 8.76m | 38.83 | 12.40% | 8.80m |
| OURS | SL+S | **425.13** | **0.21%** | 23.74m | 41.12 | 8.36% | 26.67m |

**Generalization Tests**  Finally, we study the generalization ability of discrete DIFUSCO trained on a set of TSP problems of a specific problem scale and evaluated on other problem scales. From Fig. 3, we can see that DIFUSCO has a strong generalization ability. In particular, the model trained with TSP-50 perform well on even TSP-1000 and TSP0-10000. This pattern is different from the bad generalization ability of RL-trained or SL-trained non-autoregressive methods as reported in previous work [54].

# 5  Experiments with MIS

For Maximal Independent Set (MIS), we experiment on two types of graphs that recent work [70, 1, 8, 92] shows struggles against, i.e., SATLIB [46] and Erdős-Rényi (ER) graphs [26]. The former is a set of graphs reduced from SAT instances in CNF, while the latter are random graphs. We use ER-[700-800] for evaluation, where ER-[$n$-$N$] indicates the graph contains $n$ to $N$ nodes. Following Qiu et al. [92], the pairwise connection probability $p$ is set to 0.15.

**Datasets**  The training instances of labeled by the KaMIS[3] heuristic solver. The split of test instances on SAT datasets and the random-generated ER test graphs are taken from Qiu et al. [92].

**Model Settings**  The training schedule is the same as the TSP solver (Sec. 4.1). For SATLIB, we use discrete diffusion with 50 (diffusion steps)×1 (samples) policy and 50 (diffusion steps)×4 (samples) policy as the **Greedy** and **Sampling** strategies, respectively. For ER graphs, we use continuous diffusion with 50 (diffusion steps) × 1 (samples) policy and 20 (diffusion steps) × 8 (samples) policy as the **Greedy** and **Sampling** strategies, respectively.

**Evaluation Metrics**  We report the average size of the independent set (Size), average optimality gap (Gap), and latency time (Time). The detailed description can be found in the appendix. Notice that we disable graph reduction and 2-opt local search in all models for a fair comparison since it is pointed out by [8] that all models would perform similarly with local search post-processing.

**Results and Analysis**  Tab. 3 compare discrete DIFUSCO with other baselines on SATLIB and ER-[700-800] benchmarks. We can see that DIFUSCO strongly outperforms previous state-of-the-art methods on SATLIB benchmark, reducing the gap between ground-truth and neural solvers from 0.63% to **0.21%**. However, we also found that DIFUSCO (especially with discrete diffusion in our preliminary experiments) does not perform well on the ER-[700-800] data. We hypothesize that this is because the previous methods usually use the node-based graph neural networks such as GCN [62] or GraphSage [36] as the backbone network, while we use an edge-based Anisotropic GNN (Sec. 3.4), whose inductive bias may be not suitable for ER graphs.

---

[3] https://github.com/KarlsruheMIS/KaMIS (MIT License)

# 6 Concluding Remarks

We proposed DIFUSCO, a novel graph-based diffusion model for solving NP-complete combinatorial optimization problems. We compared two variants of graph-based diffusion models: one with continuous Gaussian noise and one with discrete Bernoulli noise. We found that the discrete variant performs better than the continuous one. Moreover, we designed a `cosine` inference schedule that enhances the effectiveness of our model. DIFUSCO achieves state-of-the-art results on TSP and MIS problems, surpassing previous probabilistic NPC solvers in both accuracy and scalability.

For future work, we would like to explore the potential of DIFUSCO in solving a broader range of NPC problems, including Mixed Integer Programming (discussed in the appendix). We would also like to explore the use of equivariant graph neural networks [117, 45] for further improvement of the diffusion models on geometrical NP-complete combinatorial optimization problems such as Euclidean TSP. Finally, we are interested in utilizing (higher-order) accelerated inference techniques for diffusion model-based solvers, such as those inspired by the continuous time framework for discrete diffusion [12, 105].

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

# A Frequently Asked Questions

## A.1 Does DIFUSCO rely on high-quality solutions collected in advance?

DIFUSCO does not require exact optimal solutions to train the diffusion model-based solver. For instance, in TSP-500/1000/10000, we utilize the heuristic solver LKH-3 to generate near-optimal solutions, which is nearly as fast as the neural solvers themselves (refer to Table 2 for details). Empirically (in TSP-500/1000/10000), DIFUSCO still attains state-of-the-art performance even when trained on such near-optimal solutions rather than on exact optimal ones. The training data sizes are reported in Appendix E.

## A.2 This work simply applies discrete diffusion models to combinatorial optimization problems. Could you clarify the significance of this work in the field?

It is essential to highlight that our work introduces a novel approach to solving NP-complete problems using graph-based diffusion models. This approach has not been previously explored in the context of NP-complete problems, and we believe that the significant improvements in performance over the state-of-the-art on benchmark datasets emphasize the value of our work.

Drawing a parallel to the application of diffusion models in image generation and subsequent extension to videos, our work explores a similar expansion of the scope. While denoising diffusion models have been extensively researched and applied to image generation, DIFUSCO represents the first instance where these models are applied to NP-complete CO problems. Such novelty in problem formulation is fundamental for solving NP-complete problems instead of incremental ones from a methodological point of view.

The formulation of NP-complete problems into a discrete $\{0, 1\}$-vector space and the use of graph-based denoising diffusion models to generate high-quality solutions make our approach unique. Additionally, we explore diffusion models with Gaussian and Bernoulli noise and introduce an effective inference schedule to improve the generation quality, which further underlines the significance of our work.

In conclusion, we believe that our work makes a significant and novel contribution to the field by introducing a new graph-based diffusion framework for solving NP-complete problems. The application of diffusion models, which have been primarily used for image and video generation, to the realm of combinatorial optimization highlights the innovative aspects of our research. We hope that this clarification demonstrates its importance in the field.

## A.3 How can DIFUSCO produce diverse multimodal output distribution?

Such ability comes from the characteristics of diffusion models, which are known for generating a wide variety of distributions as generative models (like diverse images). A more comprehensive understanding can be found in the related literature, such as DDPM.

## A.4 Why is there no demonstration for the time-cost in the results shown in Table 1, and why are there some conflicts in the metrics of Length and Gaps?

The time is not reported in Table 1 as these baseline methods are evaluated on very different hardware, which makes the runtime comparison not too meaningful. As for the "conflicts", the results (both length and optimality gap) of all the baseline methods are directly copied from previous papers. The "conflicts" may be caused by rounding issues when reporting numerical numbers in previous papers, as our best guess.

The -0.01 gap in DIFUSCO is because the Concorde solver only accepts integer coordinates as the inputs, which may lead it to produce inaccurate non-optimal solutions due to rounding issues.

## A.5 How can we effectively evaluate the usefulness of the proposed method when the training time is also a consideration?

It is important to clarify that in the context of neural combinatorial optimization solvers and general machine learning research, the primary focus is typically on inference time. This is because, after

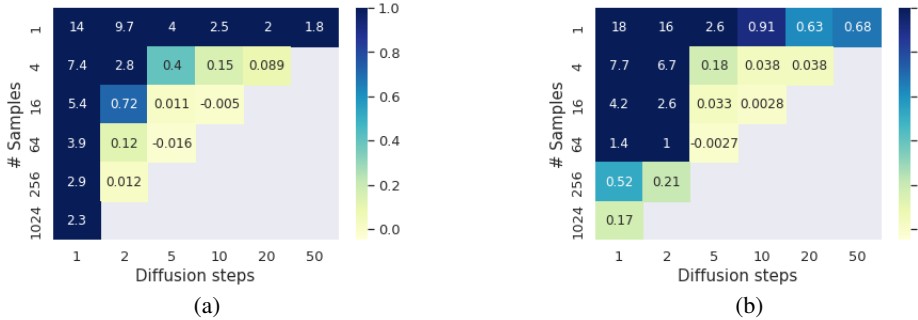

(a)                                          (b)

Figure 4: The performance Gap (%) of continuous diffusion **(a)** and discrete diffusion **(b)** models on TSP-50 with different diffusion steps and number of samples. **(c)**: The results are reported with greedy decoding without 2-opt post-processing.

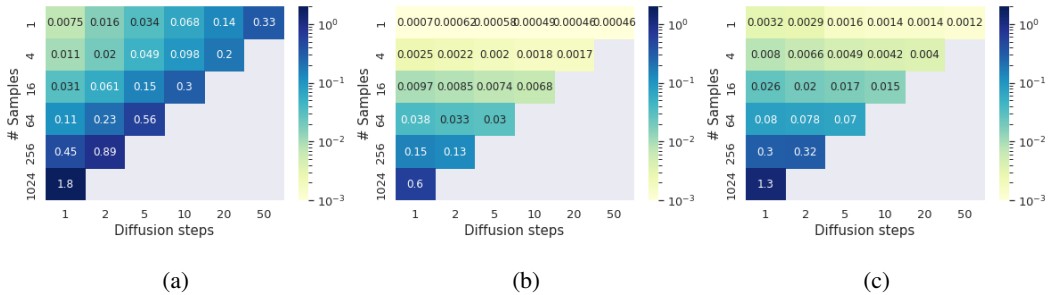

(a)                          (b)                          (c)

Figure 5: The inference per-instance run-time (sec) of diffusion models on TSP-50, where the total run-time is decomposed to neural network **(a)** + greedy decoding **(b)** + 2-opt **(c)**.

the model has been trained, it can be applied to a virtually unlimited number of unseen examples (graphs) during deployment. On the other hand, the training time is often considered negligible in comparative evaluations, partly due to the fact that traditional NCP solvers are hand-crafted instead of learning-based, and partly because the training cost for learnable models is out weighted by the benefits of their numerous applications.

### A.5.1 Can you name a real application that has many test instances and require such a model?

It is worth noting that the routing problem is a fundamental and perhaps the most important combinatorial optimization problem in real-world scenarios. One example is the food delivery problem, where a pizza shop is planning to deliver pizzas to four different addresses and then return to the store. The routing problem has significant implications for industries such as retail, quick service restaurants (QSRs), consumer packaged goods (CPG), and manufacturing. In 2020, parcel shipping exceeded 131 billion in volume globally and is projected to more than double by 2026 [91]. With the changing economic and geopolitical landscape within the transport and logistics industry, Last Mile Delivery (LMD) has become the most expensive portion of the logistics fulfillment chain, representing over 41% of overall supply chain costs [50].

## B    Extended Related Work

**Autoregressive Constructive Solvers**    Since Bello et al. [6] proposed the first autoregressive CO solver, more advanced models have been developed in the years since [22, 64, 90, 25, 68], including better network backbones [106, 64, 9]), more advanced deep reinforcement learning algorithms [57, 78, 65, 67, 87, 114, 109, 18], improved training strategies [59, 7], and for a wider range of NPC problems such as Capacitated Vehicle Routing Problem (CVRP) [83], Job Shop Scheduling Problem (JSSP) [122, 89], Maximal Independent Set (MIS) problem [57, 1, 80, 92], and boolean satisfiability problem (SAT) [119].

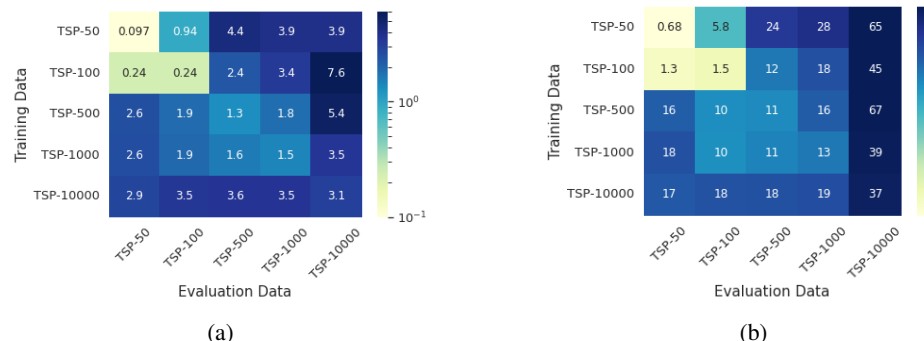

(a)                                        (b)

Figure 6: Generalization tests of discrete DIFUSCO trained and evaluated on TSP problems across various scales. The results are reported with **(a)** and without **(b)** 2-opt post-processing.

Table 4: Comparing discrete diffusion and continuous diffusion on TSP-100 with various diffusion steps and numbers of parallel sampling. `cosine` schedule is used for fast sampling.

| DIFFUSION STEPS | #SAMPLE | DISCRETE DIFFUSION (Gap%) | | CONTINUOUS DIFFUSION (Gap%) | | PER-INSTANCE RUNTIME (sec) | | |
|---|---|---|---|---|---|---|---|---|
| | | W/ 2OPT | W/O 2OPT | W/ 2OPT | W/O 2OPT | NN | GD | 2-OPT |
| 50 | 1 | 0.23869 | 1.45574 | 1.46146 | 7.66379 | 0.50633 | 0.00171 | 0.00210 |
| 100 | 1 | 0.23366 | 1.48161 | 1.32573 | 7.02117 | 1.00762 | 0.00170 | 0.00207 |
| 50 | 4 | 0.02253 | 0.09280 | 0.42741 | 1.65264 | 1.52401 | 0.00643 | 0.00575 |
| 10 | 16 | -0.01870 | 0.00519 | 0.13015 | 0.54983 | 1.12550 | 0.02581 | 0.02228 |
| 50 | 16 | -0.02322 | -0.00699 | 0.09407 | 0.30712 | 5.63712 | 0.02525 | 0.02037 |

**Improvement Heuristics Solvers**     Unlike construction heuristics, DRL-based improvement heuristics solvers use neural networks to iteratively enhance the quality of the current solution until the computational budget is exhausted. Such DRL-based improvement heuristics methods are usually inspired by classical local search algorithms such as 2-opt [19] and the large neighborhood search (LNS) [97], and have been demonstrated with outstanding results by many previous works [17, 47, 21, 20, 115, 79, 113, 76, 110, 58, 49].

Improvement heuristics methods, while showing superior performance compared to construction heuristics methods, come at the cost of increased computational time, often requiring thousands of actions even for small-scale problems with hundreds of nodes [20, 110]. This is due to the sequential application of local operations, such as 2-opt, on existing solutions, resulting in a bottleneck for latency. On the other hand, DIFUSCO has the advantage of denoising all variables in parallel, which leads to a reduction in the number of network evaluations required.

**Continuous Diffusion Models**     Diffusion models were first proposed by Sohl-Dickstein et al. [100] and recently achieved impressive success on various tasks, such as high-resolution image synthesis [23, 42], image editing [81, 95], text-to-image generation [84, 96, 94, 93, 34], waveform generation [14, 15, 72], video generation [43, 41, 99], and molecule generation [117, 45, 112].

Recent works have also drawn connections to stochastic differential equations (SDEs) [104] and ordinary differential equations (ODEs) [101] in a continuous time framework, leading to improved sampling algorithms by solving discretized SDEs/ODEs with higher-order solvers [73, 75, 74] or implicit diffusion [101].

## C    Additional Results

**Discrete Diffusion v.s. Continuous Diffusion on TSP-100**     We also compare discrete diffusion and continuous diffusion on the TSP-100 benchmark and report the results in Tab. 4. We can see that on TSP-100, discrete diffusion models still consistently outperform their continuous counterparts in various settings.

**More Diffusion Steps v.s. More Sampling (w/o 2-opt)**     Fig. 4 report the results of continuous diffusion and discrete diffusion with various diffusion steps and numbers of parallel sampling, without using 2-opt post-processing. The `cosine` denoising schedule is used for fast inference. Again, we find that discrete diffusion outperforms continuous diffusion across various settings.

Table 5: Comparison to DIMES w/ 2-opt

|  |  | TSP-500 | | TSP-1000 | | TSP-10000 | |
|---|---|---|---|---|---|---|---|
|  |  | Length | Gap | Length | Gap | Length | Gap |
| DIMES | RL+S+2OPT | 17.63 | 6.52% | 24.81 | 7.31% | 77.18 | 7.54% |
| DIMES | RL+AS+S+2OPT | 17.30 | 4.53% | 24.32 | 5.19% | 75.94 | 5.81% |
| DIFUSCO | SL+G+2OPT | 16.80 | 1.49% | 23.56 | 1.90% | 73.99 | 3.10% |
| DIFUSCO | SL+S+2OPT | 16.65 | 0.57% | 23.45 | 1.43% | 73.89 | 2.95% |

Table 6: Solution quality and computation time for learning-based methods using models trained on synthetic data (all the other baselines are trained with TSP-100) and evaluated on TSPLIB instances with 50 to 200 nodes and 2D Euclidean distances. Other baseline results are taken from Hudson et al. [49].

| Method | Kool et al. | | Joshi et al. | | O. da Costa et al. | | Hudson et al. | | Ours (TSP-50) | | Ours (TSP-100) | |
| Instance | Time (s) | Gap (%) | Time (s) | Gap (%) | Time (s) | Gap (%) | Time (s) | Gap (%) | Time (s) | Gap (%) | Time (s) | Gap (%) |
|---|---|---|---|---|---|---|---|---|---|---|---|---|
| eil51 | 0.125 | 1.628 | 3.026 | 8.339 | 28.051 | 0.067 | 10.074 | **0.000** | 0.482 | **0.000** | 0.519 | 0.117 |
| berlin52 | 0.129 | 4.169 | 3.068 | 33.225 | 31.874 | 0.449 | 10.103 | 0.142 | 0.527 | **0.000** | 0.526 | **0.000** |
| st70 | 0.200 | 1.737 | 4.037 | 24.785 | 23.964 | 0.040 | 10.053 | 0.764 | 0.663 | **0.000** | 0.670 | **0.000** |
| eil76 | 0.225 | 1.992 | 4.303 | 27.411 | 26.551 | 0.096 | 10.155 | 0.163 | 0.788 | **0.000** | 0.788 | 0.174 |
| pr76 | 0.226 | 0.816 | 4.378 | 27.793 | 39.485 | 1.228 | 10.049 | 0.039 | 0.765 | **0.000** | 0.785 | 0.187 |
| rat99 | 0.347 | 2.645 | 5.559 | 17.633 | 32.188 | 0.123 | 9.948 | 0.550 | 1.236 | 1.187 | 1.192 | **0.000** |
| kroA100 | 0.352 | 4.017 | 5.705 | 28.828 | 42.095 | 18.313 | 10.255 | 0.728 | 1.259 | 0.741 | 1.217 | **0.000** |
| kroB100 | 0.352 | 5.142 | 5.712 | 34.686 | 35.137 | 1.119 | 10.317 | 0.147 | 1.252 | 0.648 | 1.235 | **0.742** |
| kroC100 | 0.352 | 0.972 | 5.641 | 35.506 | 34.333 | 0.349 | 10.172 | 1.571 | 1.199 | 1.712 | 1.168 | **0.000** |
| kroD100 | 0.352 | 2.717 | 5.621 | 38.018 | 25.772 | 0.866 | 10.375 | 0.572 | 1.226 | **0.000** | 1.175 | 0.000 |
| kroE100 | 0.352 | 1.470 | 5.650 | 26.589 | 34.475 | 1.832 | 10.270 | 1.216 | 1.208 | 0.274 | 1.197 | **0.274** |
| rd100 | 0.352 | 3.407 | 5.737 | 50.432 | 28.963 | 0.003 | 10.125 | 0.459 | 1.191 | **0.000** | 1.172 | 0.000 |
| eil101 | 0.359 | 2.994 | 5.790 | 26.701 | 23.842 | 0.387 | 10.276 | 0.201 | 1.222 | 0.576 | 1.215 | **0.000** |
| lin105 | 0.380 | 1.739 | 5.938 | 34.902 | 39.517 | 1.867 | 10.330 | 0.606 | 1.321 | **0.000** | 1.280 | **0.000** |
| pr107 | 0.391 | 3.933 | 5.964 | 80.564 | 29.039 | 0.898 | 9.977 | 0.439 | 1.381 | **0.228** | 1.378 | 0.415 |
| pr124 | 0.499 | 3.677 | 7.059 | 70.146 | 29.570 | 10.322 | 10.360 | 0.755 | 1.803 | 0.925 | 1.782 | **0.494** |
| bier127 | 0.522 | 5.908 | 7.242 | 45.561 | 39.029 | 3.044 | 10.260 | 1.948 | 1.938 | **1.011** | 1.915 | 0.366 |
| ch130 | 0.550 | 3.182 | 7.351 | 39.090 | 34.436 | 0.709 | 10.032 | 3.519 | 1.989 | **1.970** | 1.967 | 0.077 |
| pr136 | 0.585 | 5.064 | 7.727 | 58.673 | 31.056 | 0.000 | 10.379 | 3.387 | 2.184 | **2.490** | 2.142 | 0.000 |
| pr144 | 0.638 | 7.641 | 8.132 | 55.837 | 28.913 | 1.526 | 10.276 | 3.581 | 2.478 | **0.519** | 2.446 | 0.261 |
| ch150 | 0.697 | 4.584 | 8.546 | 49.743 | 35.497 | 0.312 | 10.109 | 2.113 | 2.608 | **0.376** | 2.555 | 0.000 |
| kroA150 | 0.695 | 3.784 | 8.450 | 45.411 | 29.399 | 0.724 | 10.331 | 2.984 | 2.617 | 3.753 | 2.601 | **0.000** |
| kroB150 | 0.696 | 2.437 | 8.573 | 56.745 | 29.005 | 0.886 | 10.018 | 3.258 | 2.626 | **1.839** | 2.592 | 0.067 |
| pr152 | 0.708 | 7.494 | 8.632 | 33.925 | 29.003 | 0.029 | 10.267 | 3.119 | 2.716 | **1.751** | 2.712 | 0.481 |
| u159 | 0.764 | 7.551 | 9.012 | 38.338 | 28.961 | 0.054 | 10.428 | 1.020 | 2.963 | 3.758 | 2.892 | **0.000** |
| rat195 | 1.114 | 6.893 | 11.236 | 24.968 | 34.425 | 0.743 | 12.295 | 1.666 | 4.400 | 1.540 | 4.428 | **0.767** |
| d198 | 1.153 | 373.020 | 11.519 | 62.351 | 30.864 | 0.522 | 12.596 | 4.772 | 4.615 | 4.832 | 4.153 | **3.337** |
| kroA200 | 1.150 | 7.106 | 11.702 | 40.885 | 33.832 | 1.441 | 11.088 | 2.029 | 4.710 | 6.187 | 4.686 | **0.065** |
| kroB200 | 1.150 | 8.541 | 11.689 | 43.643 | 31.951 | 2.064 | 11.267 | 2.589 | 4.606 | 6.605 | 4.619 | **0.590** |
| Mean | 0.532 | 16.767 | 7.000 | 40.025 | 31.766 | 1.725 | 10.420 | 1.529 | 1.999 | 1.480 | 1.966 | **0.290** |

Besides, we find that without the 2-opt post-processing, performing more diffusion iterations is much more effective than sampling more solutions, even when the former uses less computation. For example, 20 (diffusion steps) $\times$ 4 (samples) not only significantly outperforms 1 (diffusion steps) $\times$ 1024 (samples), but also has a $18.5\times$ less runtime.

**Runtime Analysis** We report the decomposed runtime (neural network + greedy decoding + 2-opt) for diffusion models on TSP-50 in Fig. 5. We can see that while neural network execution takes the majority of total runtime, 2-opt also takes a non-negligible portion of the runtime, especially when only a few diffusion steps (like 1 or 2) are used.

**Generalization Tests (w/o 2opt)** We also report the generalization tests of discrete DIFUSCO without 2-opt post-processing in Fig.6 (b).

**Comparison to DIMES w/ 2opt** We compare DIMES with 2-opt to DIFUSCO in Tab. 5. We can see that while 2-opt can further improve the performance of DIMES on large-scale TPS problem instances, there is still a significant gap between DIEMS and DIFUSCO when both are equipped with 2-opt post-processing.

Table 7: Results of DIFUSCO with multiple samples for MCTS

| | TSP-500 | | TSP-1000 | | TSP-10000 | |
| | Length | Gap | Length | Gap | Length | Gap |
|---|---|---|---|---|---|---|
| PREVIOUS SOTA | 16.84 | 1.76% | 23.69 | 2.46% | 74.06 | 3.19% |
| DIFUSCO 1xMCTS | 16.64 | 0.46% | 23.39 | 1.17% | 73.62 | 2.58% |
| DIFUSCO 2xMCTS | 16.57 | 0.09% | 23.32 | 0.56% | 73.60 | 2.55% |
| DIFUSCO 4xMCTS | 16.56 | 0.02% | 23.30 | 0.46% | 73.54 | 2.48% |

**Generalization to Real-World Instances**    In many cases, there might not be enough real-world problem instances available to train a model. As a result, the model needs to be trained using synthetically generated data. Hence, it becomes crucial to assess the effectiveness of transferring knowledge from the simulated environment to the real world. we evaluated the DIFUSCO models trained on TSP-50 and TSP-100 on TSPLib graphs with 50-200 nodes and a comparison to other methods (trained TSP-100) [64, 53, 20, 49] is shown in Tab. 6.

We can see that DIFUSCO achieves significant improvements over other baselines on the real-world TSPLIB data (0.29% gap v.s. 1.48% in the best baseline). Besides, DIFUSCO also shows strong generalization ability across problem scales, where DIFUSCO trained on TSP-50 data achieves competitive performance with the best baseline Hudson et al. [49] trained on TSP-100.

**MCTS with Multiple Diffusion Samples**    One may wonder what if DIFUSCO with MCTS is allowed to have more than one sample. To evaluate the impact of using multiple samples in DIFUSCO with MCTS, we conducted experiments with 1x, 2x, and 4x greedy heatmap generation (50 diffusion steps) combined with MCTS using different random seeds, and report the results in Tab. 7.

Our findings show that DIFUSCO's performance significantly improves when MCTS decoding is applied to diverse heatmaps. We would like to highlight that the diverse heatmap + MCTS decoding approach is unique to diffusion model-based neural solvers like DIFUSCO, as previous methods (e.g., Att-GCN and DIMES) that employ MCTS only yield a unimodal distribution. While it is true that using 2x or 4x MCTS decoding would increase the runtime of DIFUSCO, we believe there is a similar exploitation versus exploration trade-off between MCTS time and MCTS seeds, akin to the trade-off between diffusion steps and the number of samples demonstrated in Figure 2 of our paper.

# D    Discussion on the $\{0, 1\}^N$ Vector Space of CO Problems

The design of the $\{0, 1\}^N$ vector space can also represent non-graph-based NP-complete combinatorial optimization problems. For example, on the more general Mixed Integer Programming (MIP) problem, we can let $\mathcal{X}_s$ be a 0/1 indication set of all extended variables[4]. $\text{cost}(\cdot)$ can be defined as a linear/quadratic function of $\mathbf{x}$, and $\text{valid}(\cdot)$ is a function validating all linear/quadratic constraints, bound constraints, and integrality constraints.

# E    Additional Experiment Details

**Metrics: TSP**    For TSP, Length is defined as the average length of the system-predicted tour for each test-set instance. Gap is the average of the relative decrease in performance compared to a baseline method. Time is the total clock time required to generate solutions for all test instances, and is presented in seconds (s), minutes (m), or hours (h).

**Metrics: MIS**    For MIS, Size (the larger, the better) is the average size of the system-predicted maximal independent set for each test-set graph, Gap and Time are defined similarly as in the TSP case.

**Hardware**    All the methods are trained with $8\times$ NVIDIA Tesla V100 Volta GPUs and evaluated on a single NVIDIA Tesla V100 Volta GPU, with $40\times$ Intel(R) Xeon(R) Gold 6248 CPUs @ 2.50GHz.

---

[4]For an integer variable $z$ that can be assigned values from a finite set with cardinality $\text{card}(z)$, any target value can be represented as a sequence of $\lceil \log_2(\text{card}(z)) \rceil$ bits [82, 16].

**Random Seeds**   Since the diffusion models can generate an arbitrary sample from its distribution even with the greedy decoding scheme, we report the averaged results across 5 different random seeds when reporting all results.

**Training Details**   All DIFUSCO models are trained with a cosine learning rate schedule starting from 2e-4 and ending at 0.

- TSP-50: We use 1502000 random instances and train DIFUSCO for 50 epochs with a batch size of 512.
- TSP-100: We use 1502000 random instances and train DIFUSCO for 50 epochs with a batch size of 256.
- TSP-500: We use 128000 random instances and train DIFUSCO for 50 epochs with a batch size of 64. We also apply curriculum learning and initialize the model from the TSP-100 checkpoint.
- TSP-1000: We use 64000 random instances and train DIFUSCO for 50 epochs with a batch size of 64. We also apply curriculum learning and initialize the model from the TSP-100 checkpoint.
- TSP-10000: We use 6400 random instances and train DIFUSCO for 50 epochs with a batch size of 8. We also apply curriculum learning and initialize the model from the TSP-500 checkpoint.
- SATLIB: We use the training split of 49500 examples from [46, 92] and train DIFUSCO for 50 epochs with a batch size of 128.
- ER-[700-800]: We use 163840 random instances and train DIFUSCO for 50 epochs with a batch size of 32.

## F   Decoding Strategies

**Greedy Decoding**   We use a simple greedy decoding scheme for diffusion models, where we sample one solution from the learned distribution $p_{\boldsymbol{\theta}}(\mathbf{x}_0)$. We compare this scheme with other autoregressive constructive solvers that also use greedy decoding.

**Sampling**   Following Kool et al. [64], we also use a sampling scheme where we sample multiple solutions in parallel (i.e., each diffusion model starts with a different noise $\mathbf{x_T}$) and report the best one.

**Monte Carlo Tree Search**   For the TSP task, we adopt a more advanced reinforcement learning-based search approach, i.e., Monte Carlo tree search (MCTS), to find high-quality solutions. In MCTS, we sample $k$-opt transformation actions guided by the heatmap generated by the diffusion model to improve the current solutions. The MCTS repeats the simulation, selection, and back-propagation steps until no improving actions are found in the sampling pool. For more details, please refer to [27].

## G   Fast Inference for Continuous and Discrete Diffusion Models

We first describe the denoising diffusion implicit models (DDIMs) [101] algorithm for accelerating inference for continuous diffusion models.

Formally, consider the forward process defined not on all the latent variables $\mathbf{x}_{1:T}$, but on a subset $\{\mathbf{x}_{\tau_1}, \ldots, \mathbf{x}_{\tau_M}\}$, where $\tau$ is an increasing sub-sequence of $[1, \ldots, T]$ with length $M$, $\mathbf{x}_{\tau_1} = 1$ and $\mathbf{x}_{\tau_M} = T$.

For continuous diffusion, the marginal can still be defined as:

$$q(\mathbf{x}_{\tau_i}|\mathbf{x}_0) := \mathcal{N}(\mathbf{x}_{\tau_i}; \sqrt{\bar{\alpha}_{\tau_i}}\mathbf{x}_0, (1 - \bar{\alpha}_{\tau_i})\mathbf{I}) \tag{9}$$

And it's (deterministic) posterior is defined by:

$$q(\mathbf{x}_{\tau_{i-1}}|\mathbf{x}_{\tau_i}, \mathbf{x}_0) := \mathcal{N}(\mathbf{x}_{\tau_{i-1}}; \sqrt{\frac{\bar{\alpha}_{\tau_{i-1}}}{\bar{\alpha}_{\tau_i}}} \left(\mathbf{x}_{\tau_i} - \sqrt{1 - \bar{\alpha}_{\tau_i}} \cdot \widetilde{\boldsymbol{\epsilon}}_{\tau_i}\right) + \sqrt{1 - \bar{\alpha}_{\tau_{i-1}}} \cdot \widetilde{\boldsymbol{\epsilon}}_{\tau_i}), \mathbf{0}) \tag{10}$$

where $\widetilde{\boldsymbol{\epsilon}}_{\tau_i} = (\mathbf{x}_{\tau_i} - \sqrt{\bar{\alpha}_{\tau_i}}\mathbf{x}_0)/\sqrt{1 - \bar{\alpha}_{\tau_i}}$ is the (predicted) diffusion noise.

Next, we describe its analogy in the discrete domain, which is first proposed by Austin et al. [5].

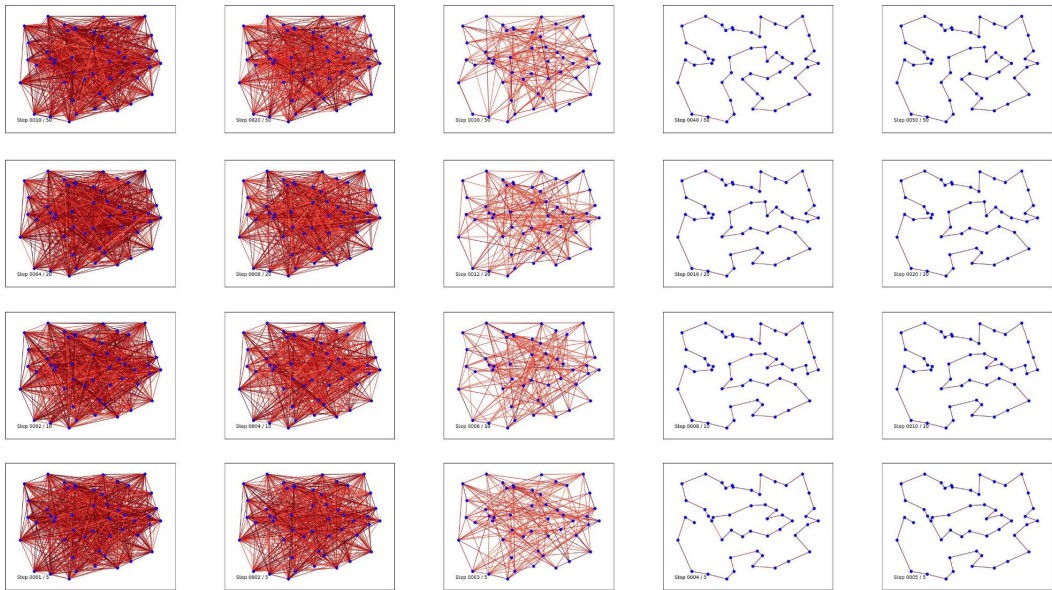

Figure 7: Qualitative illustration of how diffusion steps affect the generation quality of **continuous** diffusion models. The results are reported without any post-processing. **Continuous** DIFUSCO with 50 (first row), 20 (second row), 10 (third row), and 5 (last row) diffusion steps in `cosine` schedule are shown.

For discrete diffusion, the marginal can still be defined as:

$$q(\mathbf{x}_{\tau_i}|\mathbf{x}_0) = \text{Cat}\left(\mathbf{x}_{\tau_i}; \mathbf{p} = \mathbf{x}_0\overline{\mathbf{Q}}_{\tau_i}\right), \tag{11}$$

while the posterior becomes:

$$q(\mathbf{x}_{\tau_{i-1}}|\mathbf{x}_{\tau_i}, \mathbf{x}_0) = \frac{q(\mathbf{x}_{\tau_i}|\mathbf{x}_{\tau_{i-1}}, \mathbf{x}_0)q(\mathbf{x}_{\tau_{i-1}}|\mathbf{x}_0)}{q(\mathbf{x}_{\tau_i}|\mathbf{x}_0)}$$
$$= \text{Cat}\left(\mathbf{x}_{\tau_{i-1}}; \mathbf{p} = \frac{\mathbf{x}_{\tau_i}\overline{\mathbf{Q}}_{\tau_{i-1},\tau_i}^{\top} \odot \mathbf{x}_0\overline{\mathbf{Q}}_{\tau_{i-1}}}{\mathbf{x}_0\overline{\mathbf{Q}}_{\tau_i}\mathbf{x}_{\tau_i}^{\top}}\right), \tag{12}$$

where $\overline{\mathbf{Q}}_{t',t} = \mathbf{Q}_{t'+1}\ldots\mathbf{Q}_t$.

## H    Experiment Baselines

**TSP-50/100**    We evaluate DIFUSCO on TSP-50 and TSP-100 against 10 baselines, which belong to two categories: traditional Operations Research (OR) methods and learning-based methods.

- Traditional OR methods include Concorde [3], an exact solver, and 2-opt [71], a heuristic method.
- Learning-based methods include AM [64], GCN [53], Transformer [10], POMO [67], Sym-NCO[59], DPDP [79], Image Diffusion [32], and MDAM [114]. These are the state-of-the-art methods in recent benchmark studies.

**TSP-500/1000/10000**    We compare DIFUSCO on large-scale TSP problems with 9 baseline methods, which fall into two categories: traditional Operations Research (OR) methods and learning-based methods.

- Traditional OR methods include Concorde [3] and Gurobi [35], which are exact solvers, and LKH-3 [39], which is a strong heuristic solver. We use two settings for LKH-3: (i) *default*: 10000 trials (the default configuration of LKH-3); (ii) *less trials*: 500 trials for TSP-500/1000 and 250 trials for TSP-10000. We also include Farthest Insertion, a simple heuristic method, as a baseline.
- Learning-based methods include EAN [22], AM [64], GCN [53], POMO+EAS [67, 48], Att-GCN [27], and DIMES [92]. These are the state-of-the-art methods in recent benchmark studies. They

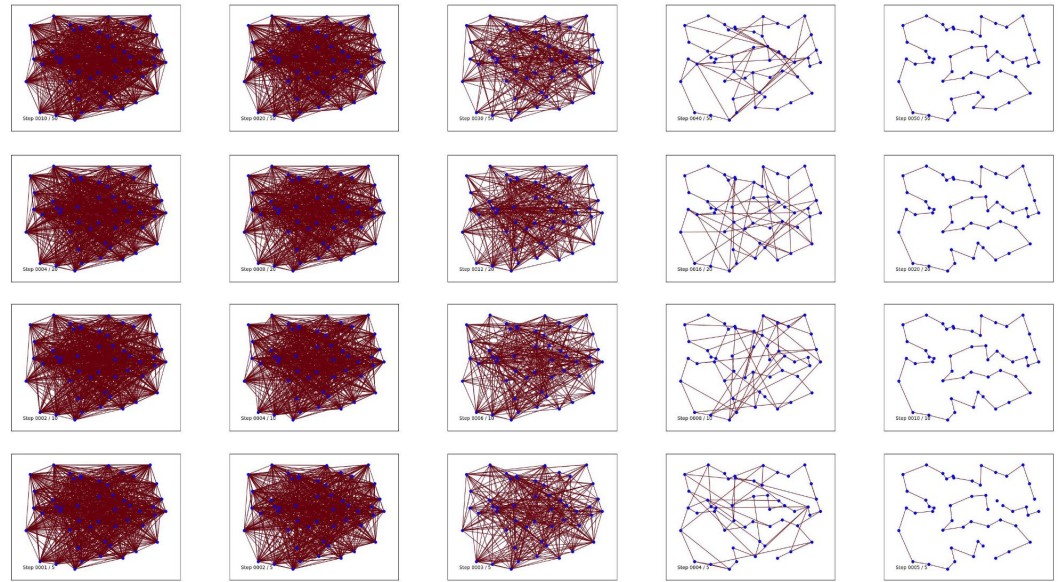

Figure 8: Qualitative illustration of how diffusion steps affect the generation quality of **discrete** diffusion models. The results are reported without any post-processing. **Discrete** DIFUSCO with 50 (first row), 20 (second row), 10 (third row), and 5 (last row) diffusion steps in `cosine` schedule are shown.

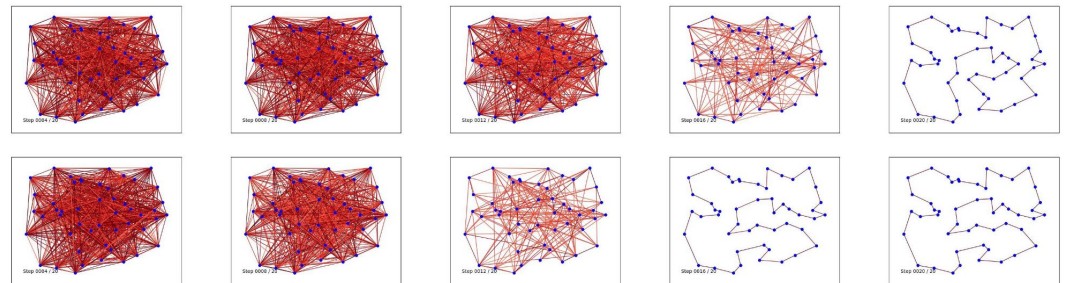

Figure 9: Qualitative illustration of how diffusion schedules affect the generation quality of **continuous** diffusion models. The results are reported without any post-processing. **Continuous** DIFUSCO with `linear` schedule (first row) and `cosine` schedule (second row) with 20 diffusion steps are shown.

can be further divided into reinforcement learning (RL) and supervised learning (SL) methods. Some RL methods can also use an Active Search (AS) stage [6] to fine-tune each instance. We take the results of the baselines from Fu et al. [27] and Qiu et al. [92]. Note that except for Att-GCN and DIMES, the baselines are trained on small graphs and tested on large graphs.

**MIS** For MIS, we compare DIFUSCO with 6 other MIS solvers on the same test sets, including two traditional OR methods (i.e., Gurobi and KaMIS) and four learning-based methods. Gurobi solves MIS as an integer linear program, while KaMIS is a heuristics solver for MIS. The four learning-based methods can be divided into the reinforcement learning (RL) category, i.e., LwD [1] and DIMES [92], and the supervised learning (SL) category, i.e., Intel [70] and DGL [8].

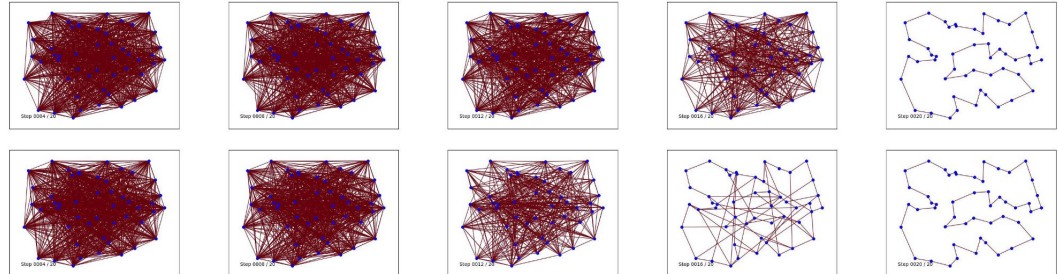

Figure 10: Qualitative illustration of how diffusion schedules affect the generation quality of **discrete** diffusion models. The results are reported without any post-processing. **Discrete** DIFUSCO with `linear` schedule (first row) and `cosine` schedule (second row) with 20 diffusion steps are shown.

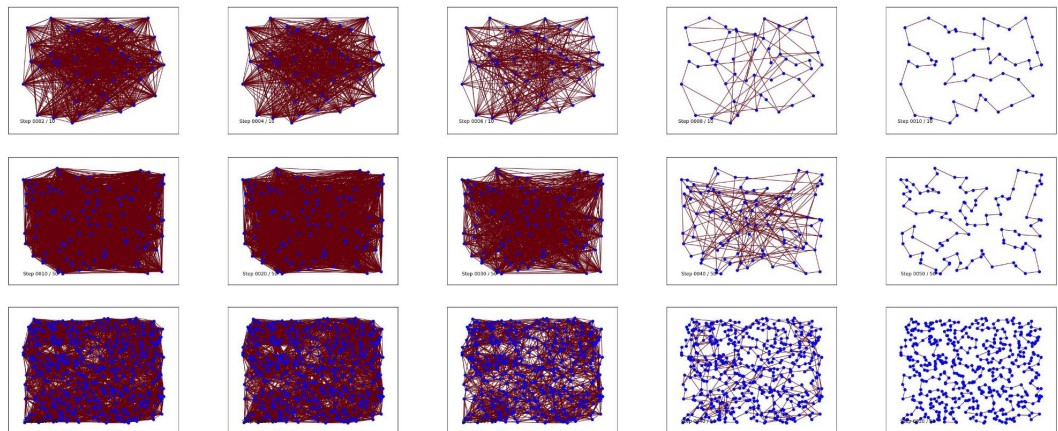

Figure 11: Qualitative illustration of discrete DIFUSCO on TSP-50, TSP-100 and TSP-500 with 50 diffusion steps and `cosine` schedule.

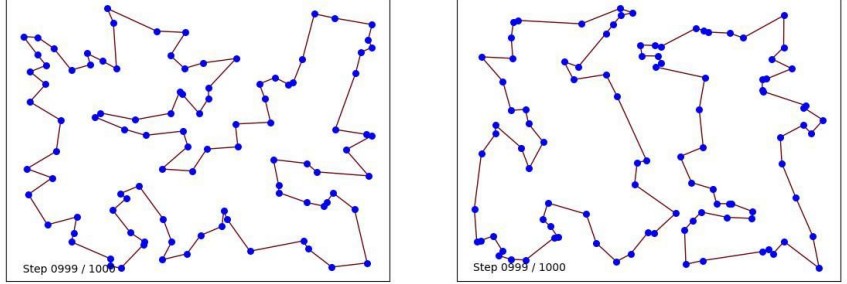

Figure 12: Success (left) and failure (right) examples on TSP-100, where the latter fails to form a single tour that visits each node exactly once. The results are reported without any post-processing.