# OpenReview forum: "DIFUSCO: Graph-based Diffusion Solvers for Combinatorial Optimization"
_NeurIPS.cc/2023/Conference — NeurIPS 2023 spotlight_

### Official Review · Reviewer_zZSi · 2023-07-03

**Soundness:** 3 good
**Presentation:** 3 good
**Contribution:** 3 good
**Rating:** 7
**Confidence:** 4

**Summary:**

This paper introduces DIFUSCO, a novel graph-based diffusion model for solving NP-complete combinatorial optimization problems. The authors propose two variants of the diffusion model: one with continuous Gaussian noise and one with discrete Bernoulli noise. Through extensive experiments on the Traveling Salesman Problem (TSP) and Maximal Independent Set (MIS) problems, they demonstrate that the discrete diffusion model outperforms the continuous one. The proposed DIFUSCO model achieves state-of-the-art results on TSP and MIS problems, surpassing previous probabilistic NPC solvers in both accuracy and scalability.

**Strengths:**

Novelty: The paper introduces a new graph-based diffusion model for solving NP-complete combinatorial optimization problems, which is a novel technique to the field.
Performance: The proposed DIFUSCO model outperforms previous state-of-the-art methods on TSP and MIS problems, demonstrating its effectiveness.The DIFUSCO model shows good scalability, able to handle larger problem instances that previous methods struggle with. The DIFUSCO model exhibits good generalization ability, with the model trained on TSP-50 performing well on even TSP-1000 and TSP-10000.


**Weaknesses:**

 The paper focuses on TSP and MIS problems, and it is unclear how well the proposed model would perform on other types of NP-complete problems.
The DIFUSCO model, especially with discrete diffusion, does not perform well on ER graphs.
The performance of the DIFUSCO model relies on 2-opt local search post-processing. Which of the model and 2-opt is the major contribution to the final performance?
The proposed DIFUSCO shares similar representation with GNN-based solvers. What are the substantial advantage over the GNN-based ones?




**Questions:**

The authors hypothesize that the poor performance due to the use of an edge-based Anisotropic GNN, whose inductive bias may not be suitable for ER graphs. Could the authors elaborate more on this issue? Are there potential modifications to the model that could improve its performance on ER graphs?
Could the authors provide more details on the performance of the model without post-processing? Furthermore, are there potential improvements to the model that could reduce its dependency on post-processing?


**Limitations:**

Societal impact of the work is not discussed in this paper. Furthermore, limitations of the proposed technique need further discussions. Maybe solving a real-world application can be helpful.

---

> ### Author Rebuttal · Authors · 2023-08-08
>
> We appreciate the reviewers taking the time to evaluate our work. Please find our responses to your concerns below:
>
> > About more Benchmark Problems
>
> We would like to cite the comments by reviewer WFTx: “It is extremely solid study in all regards.  .. The experiments are impressive in their breadth.”  We truly appreciate such an understanding.
>
> On the other hand, we must point out that solving all the NP-complete problems in a single paper is not realistic and has not been the reality in the literature of both combinatorial optimization and machine learning. We focus on the TSP and MIS problems because they are adequately representative of edge-based and node-based NP-complete CO problems, respectively, and have been commonly studied by the machine learning community for neural solvers on NP-complete problems.
>
> > Which of the model and 2-opt is the major contribution to the final performance?
>
> We've compared results with 2-opt post-processing for both DIFUSCO and DIMES in Table 2 of Appendix D: Additional Results. Evidently, DIFUSCO outperforms DIMES, even when 2-OPT is applied to both.
>
> > The proposed DIFUSCO shares similar representation with GNN-based solvers.
>
> We would like to clarify that while DIFUSCO does use GNN as a neural network backbone, it's a broader probabilistic framework for tackling graph-based NP-complete CO problems. The novelty lies in the framework, not in the neural network architecture.
>
> > Elaboration on the Inductive Bias for ER Graphs
>
> Our interpretation is that the Anisotropic GNN may not be ideal for MIS (ER) graphs because MIS is essentially a node selection task while the Anisotropic GNN is essentially designed for edge representation learning and related edge selection tasks (like TSP). Using an edge-based method to solve a node-based prediction task would unavoidably lead to wasteful computation and unnecessary complexity in modeling.  We appreciate the reviewer’s question on this and consider the in-depth investigation of this hypothesis and potential remedies as a future research topic.
>
> > Are there potential modifications to the model that could improve its performance on ER graphs?
>
> We speculate that node-based GNNs, such as GAT or GIN, as utilized in state-of-the-art neural MIS solvers [1], could potentially improve performance on ER graphs.
>
> > Could the authors provide more details on the performance of the model without post-processing?
>
> Performance data without post-processing is presented in Table 2, and a more detailed analysis is provided in Figure 3 of the appendix.
>
> > Are there potential improvements to the model that could reduce its dependency on post-processing?
>
> Our post-processing design is borrowed from prior work [2]. It would indeed be intriguing if the neural model could produce high-quality results without any post-processing. However, given that the 2-opt post-processing has been commonly used since 2018 [3] and is much more efficient than other neural or heuristic parts, we consider it as a common practice.
>
> > Societal impact of the work is not discussed in this paper.
>
> We haven't explicitly discussed societal impact as it has not been apparent yet as in the early stage of research, as pointed out by reviewers WFYx, 6WFF, and b1y4.  Nonetheless, we do believe that good learnable neural solvers for NP-complete problems will have a huge impact on a broad range of AI problems as they are the fundamental core set.
>
> > Maybe solving a real-world application can be helpful.
>
> Please note that we have already presented results on the real-world TSPLIB dataset in Appendix D: Additional Results - Generalization to Real-World Instances.
>
> ---
>
> [1] Learning what to defer for maximum in dependent sets
>
> [2] Diffusion models as plug-and-play priors
>
> [3] Learning heuristics for the TSP by policy gradient

---

> > ### Comment · Reviewer_zZSi · 2023-08-15
> > **Thank you for the response**
> >
> > Thank you for the response. I see from the results even without 2-opt the proposed model can achieve the best results in many comparisons. The other concerns have also been addressed. The introduction of diffusion model in neural solvers might represent new progress in this field. One evidence is that it is able to train on 10000-dim TSPs while most previous ones can only train on 100-dim.  I would like to raise my score from 4 to 7.

---

### Official Review · Reviewer_WFYx · 2023-07-06

**Soundness:** 4 excellent
**Presentation:** 4 excellent
**Contribution:** 3 good
**Rating:** 8
**Confidence:** 4

**Summary:**

This paper proposes and evaluates the use of diffusion models for solving combinatorial optimization problems on graphs.

The goal is to construct a probability distribution on the solution space that puts weight on high-quality solutions.
First, a dataset of optimal or near-optimal solutions is generated offline with a problem-specific solver.
Then, a diffusion model is trained in a supervised manner to maximize the likelihood of these solutions.
Finally, during inference, new solutions are generated from white noise, and put through a post-processing step to ensure feasibility.

Since many combinatorial problems exhibit a graphical structure, the diffusion model is based on a message-passing graph neural network.
To handle binary vectors, both discrete and continuous diffusion schemes are explored.
An accelerated inference scheme is suggested, along with several solution decoding and improvement heuristics.

An extensive set of benchmarks underlines the efficiency of the method compared to numerous alternatives from the state of the art.

**Strengths:**

### Originality

The main innovation here is the adaptation of diffusion models from image generation to combinatorial optimization.
It comes with a shift from convolutional to message-passing neural architectures, as well as other design questions like choosing the best type of noise process.
The combination of these known ideas is more than sufficient in terms of novelty.

### Quality

It is an extremely solid study in all regards.
The new methods are carefully justified and well-situated with respect to previous work.
The experiments are impressive in their breadth, and they do highlight the performance and promise of graph-based diffusion models.

### Clarity

I really enjoyed reading this paper.
The authors are clear and honest, they do not resort to useless mathematical details to obfuscate their main contributions.
The appendix is very thoughtfully designed, especially the FAQ section at the beginning which underlines the main motivations.

### Significance

The change of paradigm from images to graphs in diffusion models is a great foundation for future work.
While it is not a huge theoretical leap, it takes some good ideas and craftsmanship to get it right, and that's what the authors showed.
I am excited to see what comes out of it.

**Weaknesses:**

### Originality

n.a.

### Quality

n.a.

### Clarity

It would help to specify why distribution multimodality is so important for solution generation, since that seems to be a main argument in favor of diffusion models.

### Significance

The main limitation that I see is that this approach requires access to a dataset of high-quality solution, which is costly to generate.
I think the method also generalizes to unsupervised learning without much additional headache, maybe that is worth discussing as a perspective?

**Questions:**

L49: Why are non-autoregressive models limited in expressiveness?

L75: For symmetry maybe you could include related work on the third avenue from the state of the art, namely iterative improvement methods?

L83: Do all best-performing methods use attention nowadays?

L130: Would some form of regularization help in the loss?

L182: The justification of your choice of network is a bit vague, can you elaborate?

L236: Is sparsification done before or after exact solving with Concorde / LKH?

L244: I feel like tour length and drop are redundant, even though the averaging over instances makes them different. Why include both? Also, the combinatorial optimization world usually prefers "(optimality) gap" in terms of nomenclature.

L279 (Table 2): If the runtimes are not comparable, why include them?

L321: Can you elaborate on the inductive bias of your architecture for ER graphs?

Appendix L40: If the runtime is not meaningful in Table 1, why report it in Table 2?

Appendix L220: Which benchmark studies are you referring to?

**Limitations:**

Limitations are well discussed in the paper.

Societal impact is irrelevant.

---

> ### Author Rebuttal · Authors · 2023-08-08
>
> We appreciate your valuable insights and constructive feedback on our paper. Below we address your comments and suggestions:
>
> > Distribution Multimodality and Non-autoregressive Model Limitations
>
> The expressiveness of non-autoregressive models tends to be limited when confronted with multiple optimal solutions for the same graph [1]. Approaches such as multiple outputs have been suggested [1] to alleviate this issue. Our diffusion model framework presents a more principled way to address the distribution multimodality problem by iterative denoising. We will make this point more clear in the revised version of our paper.
>
> > Generalization of the current method with unsupervised learning
>
> Your suggestion in this direction is very insightful and certainly deserves exploration as future work. Our diffusion models, being probabilistic in nature [2], could potentially be trained with reinforcement learning without supervised labels in principle. However, the sparse reward issue (discussed in line 37) might pose a challenge.
>
> > Iterative Improvement Methods
>
> Due to space constraints, we have provided a detailed analysis of iterative improvement methods in our extended related work section in Appendix C.
>
> > Attention in Best-Performing Methods
>
> In autoregressive methods, the attention mechanism is essential to capture sequential order in the solution, as per previous work [3]. Variants like linear attention (Transformer as RNN) might be possible, but they have not been explored in sufficient depth in the literature so far.
>
> > Regularization in Loss
>
> Our primary goal was to keep our framework as simple as possible, and thus we did not include any additional terms in our loss function. We are also unaware of the effectiveness of regularization in diffusion models literature, which might be a good future topic to explore.
>
> > Choice of Network
>
> Our choice of network mainly follows previous work [4,5]. The Anisotropic GNN was selected because it can produce the embeddings for both nodes and edges, unlike typical GNNs such as GCN or GAT, which are designed for node embedding only. This design choice is particularly beneficial for tasks that require the prediction of edge variables like TSP. We will incorporate this clarification in the revised version of the paper.
>
> > Sparsification
>
> Sparsification was not applied to Concorde / LKH, as these solvers typically have internal mechanisms to effectively handle tasks with Euclidean distances and large-scale coordinate inputs.
>
> > Nomenclature Preferences
>
> We appreciate your advice on the usage of "gap" instead of "drop". We will incorporate this change in the revised version of the paper.
>
> > Runtime Inclusion
>
> While the runtime may not be directly comparable due to different hardware configurations and other factors, it was reported to provide readers with an understanding of our method's performance speed under different decoding strategies. We also wish to note that the runtime of traditional methods is not significantly impacted by hardware as they use CPUs.
>
> > Elaboration on the Inductive Bias for ER Graphs
>
> Our interpretation is that the Anisotropic GNN may not be ideal for MIS (ER) graphs because MIS is essentially a node selection task while the Anisotropic GNN is essentially designed for edge representation learning and related edge selection tasks (like TSP). Using an edge-based method to solve a node-based prediction task would unavoidably lead to wasteful computation and unnecessary complexity in modeling.  We appreciate the reviewer’s question on this and consider the in-depth investigation of this hypothesis and potential remedies as a future research topic.
>
> > Benchmark Studies Reference
>
> We are referring to published results reported on the popular TSP-N benchmark.
>
> ---
>
> [1]: Combinatorial Optimization with Graph Convolutional Networks and Guided Tree Search
>
> [2]: Score-Based Generative Modeling through Stochastic Differential Equations
>
> [3]: Attention, Learn to Solve Routing Problems!
>
> [4]: An experimental study of neural networks for variable graphs
>
> [5]: Dimes: A differentiable meta solver for combinatorial optimization problems

---

> > ### Comment · Reviewer_WFYx · 2023-08-14
> >
> > Thank you for taking the time to answer my questions, which have mostly been resolved.
> >
> > > We also wish to note that the runtime of traditional methods is not significantly impacted by hardware as they use CPUs.
> >
> > This is the only part that doesn't convince me at all, especially if the original implementations involve multithreading (which I don't know). I concede it is fairly minor though.

---

### Official Review · Reviewer_NgPq · 2023-07-06

**Soundness:** 3 good
**Presentation:** 3 good
**Contribution:** 2 fair
**Rating:** 6
**Confidence:** 3

**Summary:**

The paper, "DIFUSCO: Graph-based Diffusion Solvers for Combinatorial Optimization," introduces a novel methodology for solving NP-Complete combinatorial optimization problems using graph-based diffusion models, named DIFUSCO. By formulating each NP-Complete problem as a {0, 1}-valued vector with N variables that represent node or edge selection in candidate solutions, and employing a message passing-based graph neural network for instance graph encoding and variable denoising, DIFUSCO addresses the limitations of existing solution generation methods. Specifically, it performs parallel inference with fewer denoising steps, models a multimodal distribution via iterative refinements, and uses efficient, stable supervised denoising for training. The authors show DIFUSCO's superior performance over previous probabilistic solvers for two NP-complete problems, the Traveling Salesman Problem and Maximal Independent Set, using benchmark datasets.

**Strengths:**

Novel Approach: The paper introduces DIFUSCO, a novel approach that combines discrete diffusion models with improved fast unfolding of communities for combinatorial optimization problems. This new methodology could potentially open up new research directions in the field.

Thorough Experiments: The authors have conducted comprehensive experiments to validate the performance of DIFUSCO, considering different model configurations and problem scales. The experiments and the comparison with existing methods are rigorous and well-documented.

Balancing Exploration and Exploitation: The authors consider the trade-off between the number of diffusion steps and the number of samples, which is a critical aspect in the field of optimization.

**Weaknesses:**

Computational results: the numerical results on the benchmark problems don't show very significant improvement. The traditional heuristic method still performs better.

Complexity: The DIFUSCO model and its components, such as the discrete diffusion and the unfolding of communities, can be quite complex. This might make it difficult for some practitioners to fully understand and implement the method.

Dependency on External Solvers: The model relies on external solvers (like Concorde and LKH-3) for generating and labeling training instances, which might limit its applicability in settings where these solvers are not available or not preferred.

Limited Benchmark Problems: While the TSP is a widely used benchmark, it would be helpful to see how DIFUSCO performs on other combinatorial optimization problems to demonstrate its generality.

Computationally Expensive: The method seems to require substantial computational resources, especially for large-scale problems, which could limit its practical application in resource-constrained settings.

Lack of Analysis on Hyperparameters: There seems to be a limited discussion about how different hyperparameters, such as the number of diffusion steps or the noise schedule, affect performance. Some more insights into these aspects would enhance the utility of the proposed method.

**Questions:**

see the weakness

---

> ### Author Rebuttal · Authors · 2023-08-08
>
> We appreciate Reviewer NgPq for their time and feedback on our work. Let's address your concerns:
>
> > The traditional heuristic method still performs better.
>
> It's critical to highlight that our proposed method, in its simplicity, surpasses other neural solvers consistently. To date, there has been no singular neural solver able to robustly outperform heuristic solvers across the board, which sets a realistic benchmark for performance expectations.
>
> > The diffusion models are quite complex
>
> We would like to argue that diffusion models present an elegant probabilistic framework. With their proven success in other domains like image generation, we are proud to show its applicability and success in solving NP-complete problems in this paper, just as pointed out by reviewer WFYx and others.  We do believe that such cutting-edge machine learning algorithms can be mastered by ML practitioners. Our open-source code should further facilitate the understanding and adoption of our methodology.
>
> > The model relies on external solvers (like Concorde and LKH-3)
>
> We have discussed this in Appendix B.1 of the Frequently Asked Questions section. We acknowledge the potential limitation where external solvers may not be available or preferred. We would be more than willing to discuss such issues for concrete cases when they occur. However, to our knowledge so far, we haven't encountered such situations.
>
> > About more Benchmark Problems
>
> We would like to cite the comments by reviewer WFTx: “It is extremely solid study in all regards.  .. The experiments are impressive in their breadth.”  We truly appreciate such an understanding.
>
> On the other hand, we must point out that solving all the NP-complete problems in a single paper is not realistic and has not been the reality in the literature of both combinatorial optimization and machine learning. We focus on the TSP and MIS problems because they are adequately representative of edge-based and node-based NP-complete CO problems, respectively, and have been commonly studied by the machine learning community for neural solvers on NP-complete problems.
>
> > It could limit its practical application in resource-constrained settings.
>
> We would argue that large-scale neural network models have been already successfully applied to many real-world problems, as the most impressive evidence for the success of AI in the past ten years.  There is no reason for us to believe that diffusion models would be an exception. In fact, one of the advantages of neural diffusion models is their capability to scale, as parallel computing is naturally suitable for diffusion models.
>
> > Lack of Analysis on Hyperparameters:
>
> Please find our detailed analysis in Figure 2 and on lines 260 - 264. Please let us know if there are specific aspects you feel overlooked.
>
> > About "unfolding of communities" in your summary of our approach
>
> Would you kindly clarify what you mean by this phrase and why it is relevant to our paper?

---

> ### Comment · Area_Chair_QRgW · 2023-08-08
>
> Dear referee
>
> Thank you for your report. The author(s) are wondering what "unfolding of communities" relates to -- could you please clarify?
>
> Many thanks

---

> ### Author Response · Authors · 2023-08-17
> **Has our rebuttal addressed your concerns?**
>
> Dear Reviewer NgPq,
>
> We noticed that we have yet to receive a response to our rebuttal or to the AC's question. May we inquire if our rebuttal has sufficiently addressed your concerns? Please let us know if you need further explanations or clarifications. We sincerely value and appreciate your time and efforts in reviewing our work.
>
> Warm regards,
> Authors

---

> > ### Comment · Reviewer_NgPq · 2023-08-18
> >
> > Thanks for the authors' reply. After reading the rebuttal and paper one more time. I think I have misunderstood parts of the paper. I agree with the other reviewers' comments and have increased my score.

---

### Official Review · Reviewer_6WFF · 2023-07-10

**Soundness:** 3 good
**Presentation:** 3 good
**Contribution:** 3 good
**Rating:** 6
**Confidence:** 5

**Summary:**

This paper proposes a new, non-autoregressive, approach to Neural Combinatorial Optimization. As in similar non-autoregressive NCO works, a heat map characterizing the likelihood of a node being part of a solution is generated. The novelty is in the use of a generative model based on diffusion, trained in a supervised way on a sample of problem instances with pre-computed solutions as ground-truth. Two standard variants of diffusion, discrete and continuous, are proposed and experimented, as well as a (standard) trick to speed up test time diffusion. Performance comparisons with a number of NCO baselines are provided, both when test and training data are from the same distribution and when test instances are an order of magnitude larger than the training ones. The approach is tested on two distinct CO problems: TSP and MIS.


**Strengths:**

1. The paper presents a novel diffusion approach for CO, that nicely extends a previous work that was based on image diffusion [22], to better represent the generation process for CO problem solving
1. The resulting proposal is fairly original, clear, and of good quality.
1. The fact that two distinct problems can be addressed by the same architecture is a favorable indicator of the breadth of the method.
1. The comparison with prior work is fairly extensive.



**Weaknesses:**

1. The performance on large TSP instances seems to significantly rely on the 2-opt heuristic. Indeed the addition of 2-opt allows to gain about one order of magnitude in the results of DIFUSCO: Table 2 -- e.g. for TSP10000, the opt gap drops from more than 30% with a greedy or sampling strategy to about 3% after applying a 2-opt.
1. In general the use of 2-opt in the greedy decoding (line 242) considerably biases the results of Table 2. The performance of any of the compared models would likely also be boosted if the same 2-opt were applied at the end.
1. Autoregressive models are criticized for the "quadratic complexity" in their attention mechanism (line 83). One (standard) way to mitigate it is by limiting a priori (using a mask) the number of tokens allowed to influence a token. DIFUSCO is presented as a solution to this complexity bottleneck, but it actually faces the same problem in another form and uses a similar solution (Sec 4.1, paragraph "Graph sparsification", line 238).
1. Section 3.2 and Eqs. (4)-(8) are essentially a copycat of the cited sources, in particular [5,12,28]. It is too detailed for cursory reading, but not detailed enough to really understand without going back to the source. A Bayesian network diagram specifying the class of conditional probability of each variable could be more clear.



**Questions:**

1. Could the authors motivate the formula of line 211 to rank the edges. Is it likely to apply to other CO problems beyond TSP?

1. In Table 1, how is the optimality gap computed exactly? The reported value of -0.01% implies that the solutions are better than the optimal solutions returned by Concorde.


**Limitations:**

Technical limitations are not explicitly mentioned. Societal impact is not discussed either, but none is apparent, beyond the non-specific impact due to being part of AI research.

---

> ### Author Rebuttal · Authors · 2023-08-08
>
> We express our sincere gratitude to the reviewer for providing us with valuable insights on our paper. We would like to address your concerns as follows:
>
> > The performance on large TSP instances seems to significantly rely on the 2-opt heuristic
>
> We value your suggestion on the application of the 2-opt heuristic. We'd like to note that we have included results with 2-opt post-processing for both DIFUSCO and DIMES in Table 2 of Appendix D: Additional Results. The results indicate that DIFUSCO continues to outperform DIMES, even when 2-OPT is applied.
>
> > About the quadratic complexity of autoregressive models
>
> We would like to point out that there is no evidence in the neural CO literature suggesting that autoregressive models can work well with a limited context window.  Thus, the quadratic complexity is generically true for those models. Furthermore. their autoregressive generation nature slows down the inference, as supported by the literature on non-autoregressive sequence generation [1, 2].
>
> > Section 3.2 and Eqs. (4)-(8) are too detailed for cursory reading, but not detailed enough to really understand. A Bayesian network diagram could be more clear.
>
> Our intent with the section in question was to provide a detailed yet concise description of the diffusion models. As noted by Reviewer b1y4, "The authors very clearly explain the presented methods, which is quite impressive given the space limitations." Our writing logic follows this structure:
>
> 1. VAE (Bayesian) interpretation of diffusion models
> 2. Marginal distribution, posterior, and predicted posterior of discrete diffusion
> 3. Marginal distribution, posterior, and predicted posterior of continuous diffusion for discrete data
>
> > Could the authors motivate the formula of line 211 to rank the edges.
>
> The formula we used to rank the edges is taken directly from previous work [3] in order to maintain a fair comparison. Our interpretation of this formula from [3] is that it tries to convert a weighted directed graph into a weighted undirected graph with a distance heuristic.
>
> > In Table 1, how is the optimality gap computed exactly?
>
> Please refer to Appendix B.4 in the Frequently Asked Questions section for a detailed explanation of how the optimality gap was calculated.
>
> ---
>
> [1]: Non-Autoregressive Neural Machine Translation
>
> [2]: Step-unrolled Denoising Autoencoders
>
> [3]: Diffusion models as plug-and-play priors

---

> > ### Comment · Reviewer_6WFF · 2023-08-21
> > **Thanks for the rebuttal**
> >
> > I thank the authors for their concise answers.
> > After reading the rebuttal and the other reviews, I confirm that I remain supportive of the paper acceptance.

---

### Official Review · Reviewer_b1y4 · 2023-07-26

**Soundness:** 3 good
**Presentation:** 4 excellent
**Contribution:** 3 good
**Rating:** 7
**Confidence:** 3

**Summary:**

This paper investigates using diffusion models to solve combinatorial optimization problems. The authors propose two alternative methods to adapt diffusion models to this task: (1) a discrete diffusion model with Bernoulli noise, and (2) a more standard continuous diffusion model, where post-processing is needed to produce a discrete prediction.

**Strengths:**

1.	The paper is well-written. My background is more in combinatorial optimization, and I am familiar with standard diffusion models. The authors very clearly explain the presented methods, which is quite impressive given the space limitations.
2.	The methodologies appear sound, and the design decisions are presented with accompanying rationale.
3.	The computational experiments are very thorough and show the proposed algorithms give high quality solutions in low computational times. The results are discussed clearly.


**Weaknesses:**

1.	The feasibility of a solution is only treated as a Boolean (zero or infinity in eq. 1). This removes information that could be given to the solver regarding how close to feasible a particular solution is.
2.	It is slightly that most implementation details (sections 3.3, 3.4, etc.) are in the appendix. It would be nice to give some summaries in the main text, especially of key components such as decoding strategies, but I understand this may not be possible given the space limitations.
3.	The solution generalizability is slightly limited. In combinatorial optimization, the best solution strategy often changes depending on the graph (problem) sparsity. It would be a nice result to investigate sensitivity to problem sparsity for TS or pairwise connection probability for MIS.


**Questions:**

1.	Solution times are missing from Table 1.

**Limitations:**

Not presented, but not particularly relevant.

---

> ### Author Rebuttal · Authors · 2023-08-08
>
> We appreciate the time and effort the reviewer put into examining our work and providing valuable feedback. We would like to address your concerns as follows:
>
> > The feasibility of a solution is only treated as a Boolean.
>
> In our current design, feasibility is treated as a binary-valued variable. This is because assigning a continuous score to represent the feasibility of a solution for complex CO problems like TSP remains an open challenge in the field and is beyond our current focus. However, we agree with the reviewer about the potential benefits of such a treatment (using continuous scores), and further investigation on that topic is a meaningful direction.
>
> > Most implementation details are in the appendix.
>
> We are grateful for your understanding regarding the space limitations. In the revised version of our paper (with an additional page), we aim to provide more summary-level insights into key components such as decoding strategies within the main text.
>
> > Solution generalizability.
>
> Thank you for pointing out this aspect. Please notice that in Figure 3 of our paper, we explored the solution generalizability across different scales and levels of sparsity. We tested our model on real-world TSPLIB graphs, and the results are presented in Table 3 in the Appendix. We hope those are satisfactory for this concern.
>
> > Solution times are missing from Table 1.
>
> We appreciate your attention to this detail. Please refer to Appendix B.4 in the Frequently Asked Questions section, where we have addressed this issue. In short, the runtime may vary significantly based on hardware configuration and other factors, hence was not included in Table 1.

---

> > ### Comment · Reviewer_b1y4 · 2023-08-18
> >
> > Thank you for responding to my questions.
> >
> > I still believe the solution times for Table 1 should be provided, perhaps along with Appendix B.4. While the authors are correct that differing hardware make an algorithmic comparison less meaningful, there is a strong practical relevance in knowing the (hardware-dependent) computational expense for different solution methods for CO problems. This could be presented with the existing discussion about differing hardware, and rationale for this.
> >
> > My positive impression of the paper remains unchanged.

---

### Comment · Area_Chair_QRgW · 2023-08-18

I would like to thank the authors for providing detailed responses to all referee reports, and I apologize that some of the referees have not responded to the rebuttals despite multiple reminders. I will account for that in my final recommendation.

---

### Decision · Program_Chairs · 2023-09-21

**Decision:**

Accept (spotlight)

**Comment:**

Overall, I agree with the review team that this is a strong paper that should be accepted at NeurIPS. I am grateful to the authors that they have responded to all of the referees' comments in detail. I agree with two of the referees that runtime information would be useful to add to the paper, but ultimately I will leave this to the authors.

Thank you for submitting your work to NeurIPS!